# Transcriptional enhancers in human neuronal differentiation provide clues to neuronal disorders

Masahito Yoshihara [1,2,3,4,13], Andrea Coschiera [1,13], Jörg A Bachmann [5,13], Mariangela Pucci [1,6], Haonan Li [1], Shruti Bhagat [7], Yasuhiro Murakawa [7,8,9,10], Jere Weltner [11,12], Eeva-Mari Jouhilahti [11,12], Peter Swoboda [1 ✉], Pelin Sahlén [5 ✉] & Juha Kere [1,11,12 ✉]

## Abstract

Genome-wide association studies (GWASs) have identified thousands of variants associated with complex phenotypes, including neuropsychiatric disorders. To better understand their pathogenesis, it is necessary to identify the functional roles of these variants, which are largely located in non-coding DNA regions. Here, we employ a human mesencephalic neuronal cell differentiation model, LUHMES, with sensitive and high-resolution methods to discover enhancers (NET-CAGE), perform DNA conformation analysis (Capture Hi-C) to link enhancers to their target genes, and finally validate selected interactions. We expand the number of known enhancers active in differentiating human LUHMES neurons to 47,350, and find overlap with GWAS variants for Parkinson's disease and schizophrenia. Our findings reveal a fine-tuned regulation of human neuronal differentiation, even between adjacent developmental stages; provide a valuable resource for further studies on neuronal development, regulation, and disorders; and emphasize the importance of exploring the vast regulatory potential of non-coding DNA and enhancers.

**Keywords** CAGE; Enhancer; Hi-C; LUHMES; Neuron
**Subject Categories** Chromatin, Transcription & Genomics; Molecular Biology of Disease; Neuroscience

## Introduction

The regulation of gene expression is now understood to depend largely on cis-acting elements, including transcriptional enhancers, many of which are located in poorly annotated intergenic regions. Enhancers have been identified in numerous cell types, and data indicate that enhancers are highly cell-type-specific and even developmental stage-specific (Andersson et al, 2014; Arner et al, 2015; Farh et al, 2015; Hirabayashi et al, 2019). Therefore, there is a need to identify enhancers in a more detailed manner using novel experimental approaches.

Neuropsychiatric disorders comprise a large group of medical conditions ranging from relatively mild (such as developmental dyslexia (Kere, 2014)) to severe (such as schizophrenia (Sullivan and Geschwind, 2019)). These conditions have received major attention in genome-wide association studies (GWASs). The NHGRI-EBI GWAS Catalog lists ~25,000 single nucleotide polymorphism (SNP) associations from ~2500 studies on "nervous system disease" (Buniello et al, 2019). As most disease-associated variants are located within non-coding regions with enhancer potential (Ernst et al, 2011; Maurano et al, 2012), there is a need to explore the enhancer landscape in cells relevant to neuropsychiatric disorders.

Many of these disorders may have their origins in nervous system development; thus, we sought to identify uncharacterized enhancers using a comprehensive, genome-wide approach involving the Lund human mesencephalic (LUHMES) cell line, derived from human mesencephalic neuronal stem cells of a female fetus. LUHMES cells are non-malignant, possess a normal 46,XX karyotype, and have been immortalized with a *MYC* transgene, allowing for their facile propagation in vitro (Shah et al, 2016). LUHMES neuronal precursor cells can be differentiated into mature dopaminergic neurons within 6–7 days in culture. We have previously shown that differentiated LUHMES neurons are functional and exhibit high transcriptional similarity to human midbrain-derived dopaminergic neurons. (Coschiera et al, 2023; Coschiera et al, 2024; Lauter et al, 2020).

Cap analysis of gene expression (CAGE) is a method that captures the transcription start sites (TSSs) at high nucleotide resolution. This technique enabled the identification of 184,827 promoters in a variety of human cells and tissues in the FANTOM5 project (Consortium et al, 2014). CAGE also identified 65,423 enhancers by detecting enhancer RNAs (eRNAs) transcribed from active enhancers (Andersson et al, 2014; Arner et al, 2015). It has

[1]Department of Medicine Huddinge (MedH), Biosciences and Nutrition Unit, Karolinska Institutet, Stockholm, Sweden. [2]Institute for Advanced Academic Research, Chiba University, Chiba, Japan. [3]Department of Artificial Intelligence Medicine, Graduate School of Medicine, Chiba University, Chiba, Japan. [4]Premium Research Institute for Human Metaverse Medicine (WPI-PRIMe), Osaka University, Suita, Osaka, Japan. [5]Science for Life Laboratory, KTH - Royal Institute of Technology, Stockholm, Sweden. [6]Department of Bioscience and Technology for Food, Agriculture and Environment, University of Teramo, Teramo, Italy. [7]Institute for the Advanced Study of Human Biology, Kyoto University, Kyoto, Japan. [8]RIKEN-IFOM Joint Laboratory for Cancer Genomics, RIKEN Center for Integrative Medical Sciences, Yokohama, Japan. [9]IFOM - the FIRC Institute of Molecular Oncology, Milan, Italy. [10]Department of Medical Systems Genomics, Graduate School of Medicine, Kyoto University, Kyoto, Japan. [11]Folkhälsan Research Centre, Helsinki, Finland. [12]Stem Cells and Metabolism Research Program, University of Helsinki, Helsinki, Finland. [13]These authors contributed equally: Masahito Yoshihara, Andrea Coschiera, Jörg A Bachmann. ✉E-mail: peter.swoboda@ki.se; pelinak@kth.se; juha.kere@ki.se

been demonstrated that transcribed enhancers are much more likely to be functional than untranscribed enhancers identified by histone modifications or DNase I hypersensitive sites (Andersson et al, 2014). However, eRNAs are difficult to detect due to their susceptibility to degradation and low expression levels. To overcome this problem, a modified method to enrich nascent RNA, native elongating transcript (NET)-CAGE, has recently been developed (Hirabayashi et al, 2019). Using just five human cell lines, NET-CAGE identified 20,363 novel enhancers, suggesting that many more, specific for different tissues, remain unidentified.

Linking enhancers to their target genes is challenging because a single enhancer can regulate multiple genes, and an enhancer does not always regulate the nearest gene (Murakawa et al, 2016). Targeted chromosome conformation capture (Capture Hi-C or HiCap) is a powerful method for identifying enhancer–promoter interactions by combining Hi-C technology with targeted sequencing (Sahlén et al, 2015). Capture Hi-C can identify target genes of enhancers that are located tens to hundreds of kilobases (kb) away, about 67% of which are not the nearest genes (Åkerborg et al, 2019; Sahlén et al, 2021; Song et al, 2019).

Here, we first applied the NET-CAGE approach to identify enhancers at three stages of the LUHMES neuronal differentiation time course and looked for matching enhancer-GWAS hit associations based on a list of neuronal disorder associations. We selected a set of identified LUHMES enhancers and a number of SNPs implicated in neuronal disorders, and designed a Capture Hi-C probe set targeting promoters and these selected regions to map promoters and possibly other genomic regions that interact with the selected sites.

In summary, we identified 47,350 active putative enhancers in differentiating human LUHMES neurons, of which 31,057 (65.6%) are newly identified. Our analyses show the enrichment of active putative enhancers among GWAS peaks for neuronal disorders, especially Parkinson's disease and schizophrenia. By leveraging Capture Hi-C data, we show that only 41% of the enhancers interact with their nearest gene, and 32.4% of the differentially expressed genes connect to at least one differentially expressed enhancer across neuronal differentiation. Our results nearly triple the number of known active enhancers in human LUHMES neurons, reinforce the importance of enhancers as key drivers of GWAS findings, and provide a rich source of information for further studies on the pathogenesis of Parkinson's disease, schizophrenia, and other neuropsychiatric disorders (Bloem et al, 2021; Falk et al, 2016; Sullivan and Geschwind, 2019).

# Results

## Promoter and enhancer activities in differentiating human LUHMES neurons

To monitor the transcriptional changes during neuronal differentiation at the promoter and enhancer levels, we applied CAGE and NET-CAGE approaches to LUHMES cells and neurons at three differentiation stages (Fig. 1A): early differentiation (Day 1; transition from precursor cells to differentiating neurons; the beginning of ciliation, and axon outgrowth, and initiation of axon branching), mid-differentiation (Day 3; maximum ciliation; continued, active axon outgrowth and branching), and late

differentiation/neuronal maturation (Day 6; cilia disassembly; synapses have started to form, with formation of neuronal networks) (Coschiera et al, 2023; Coschiera et al, 2024; Lauter et al, 2020). The transcriptional activities of promoters and enhancers were captured by CAGE and NET-CAGE using total and nascent RNA, respectively. NET-CAGE detected 36,446 bidirectionally transcribed loci, of which 3890 (10.7%) overlapped with enhancers previously identified in the FANTOM5 project (Andersson et al, 2014; Arner et al, 2015) or the NET-CAGE development study (NET_dev) (Hirabayashi et al, 2019) (Fig. 1B). This finding indicates that the total number of transcribed enhancers reached 118,342 when the newly identified putative enhancers were combined with all previously identified enhancers (Fig. 1C). After counting the reads mapped to these enhancer regions and filtering out those with low expression, 47,350 putative enhancers were retained (Dataset EV1). Of these, 31,057 (65.6%) were novel (Fig. 1D). Given that the FANTOM5 dataset did not include the differentiating LUHMES neuronal cell line, these novel putative enhancers might be specific to neuronal progenitors and play a crucial role in neuronal differentiation. The remaining known enhancers showed the highest overlap with the brain-specific enhancers, confirming the validity of this study (Fig. EV1).

Promoter expression was measured using the 184,827 promoter regions identified in the FANTOM5 project. Among them, 52,076 were expressed in at least one LUHMES differentiation sample. Notably, 21,907 of these were temporally differentially expressed, indicating dynamic changes in gene expression during neuronal differentiation. These differentially expressed promoters were classified into four clusters based on their expression patterns (Dataset EV2). Promoters in Cluster 1 were downregulated from Day 3, whereas those in Cluster 2 were drastically downregulated on Day 6 (Fig. 2A). These downregulated promoters were enriched with genes involved in aerobic respiration, ribosome biogenesis, or splicing, possibly due to the transition of the cell state from proliferating progenitors to differentiating neurons. In contrast, promoters in Cluster 3 were upregulated from Day 3, while those in Cluster 4 were drastically upregulated on Day 6. These upregulated promoters were enriched with genes related to neuron or axon development (Fig. 2B).

Of the 47,350 putative enhancers identified, 2681 were temporally differentially expressed. Because enhancer expression levels were much lower than those of promoters, the number of differentially expressed enhancers was smaller. These differentially expressed enhancers were also classified into four clusters (Dataset EV3), similar to the promoters (Fig. 2C). Next, we examined the clusters to which the promoters nearest to the enhancers in each cluster belonged. Although the nearest promoters to the enhancers were most frequently found in the same cluster, the proportion was only 9.7–17.8%, suggesting that many enhancers target genes that are not the nearest to them (Fig. 2D). This observation highlights the importance of using additional methods to identify the target genes of enhancers.

## Identification of enhancer-associated sequence motifs provides clues to transcription factor (TF) function

The transcriptional regulation of promoters and enhancers is mediated by the binding of TFs. To elucidate the specific TFs that regulate the activity of promoters and enhancers during neuronal

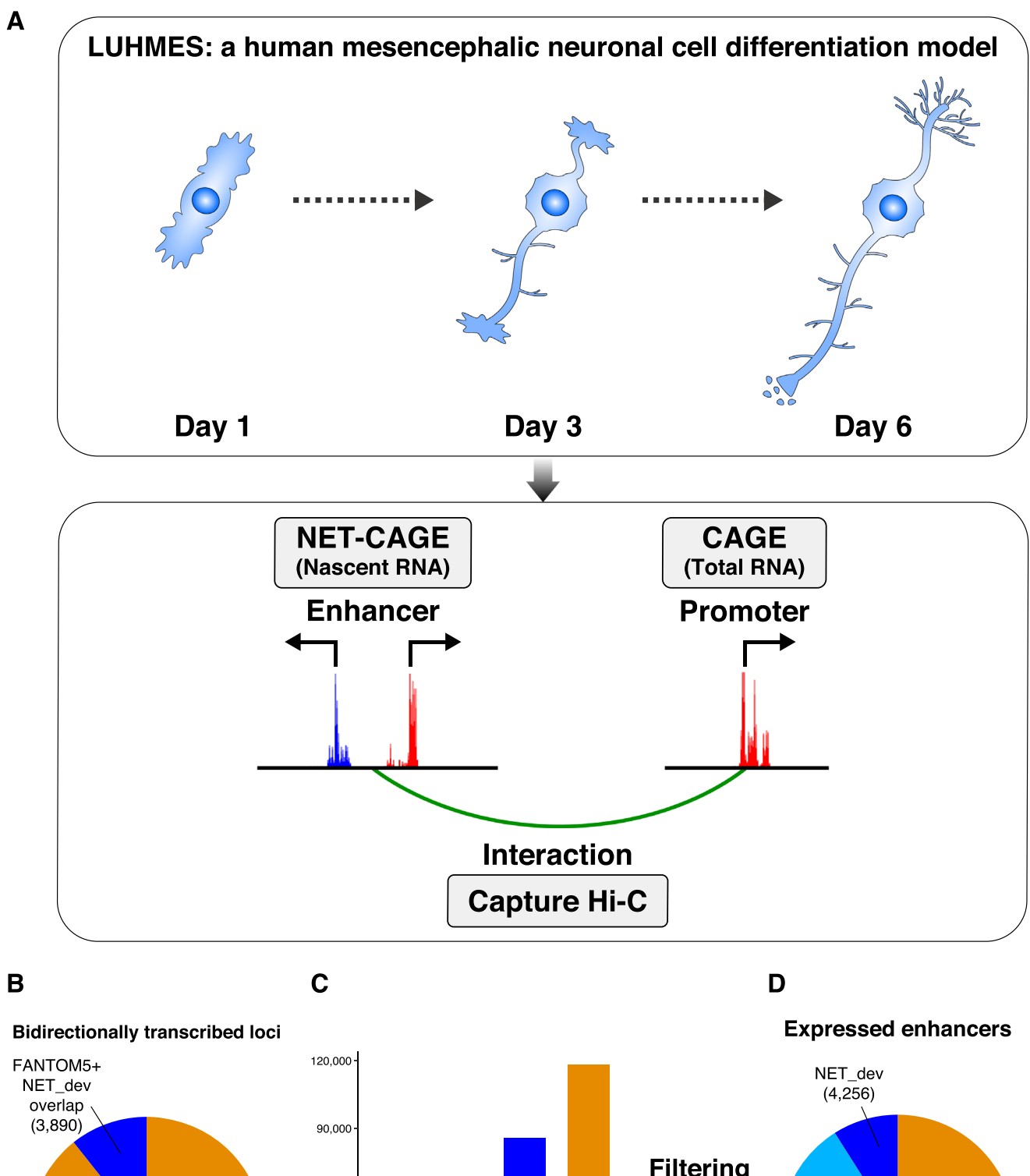

◀  **Figure 1.  Study design and identification of putative enhancers in LUHMES.**

(**A**) Study design. The human neuronal cell model LUHMES was used to identify promoters by CAGE, putative enhancers by NET-CAGE, and their interactions by Capture Hi-C. (**B**) Pie chart classifying bidirectionally transcribed loci. In total, 3890 of them overlapped with loci previously identified by the FANTOM5 project or the NET-CAGE development paper (NET_dev). (**C**) Bar plot showing the numbers of enhancers identified in FANTOM5 phases 1 and 2, NET_dev, and all enhancers combined (FANTOM5 + NET_dev + this study). (**D**) Pie chart showing the classification of the putative enhancers expressed in LUHMES after filtering out minimally expressed enhancers. Enhancers with $\log_2 [\text{CPM}] > -2.5$ in at least one sample were retained. As a result, 31,057 of these enhancers were newly identified.

differentiation, we performed TF binding motif enrichment analysis on the promoters and enhancers in each cluster. The binding motif of NFY, a positive regulator of cell proliferation, was enriched in the downregulated promoters in Clusters 1 and 2, indicating that proliferation-related genes are downregulated (Fig. 3A). Promoters of *NFYA* and *NFYC* also belonged to these downregulated clusters (Fig. EV2A). The binding motif of NeuroD1, previously shown to convert fibroblasts to neuronal cells (Pang et al, 2011), was enriched in the downregulated enhancers in Clusters 1 and 2 (Fig. 3B). It was also the most enriched binding motif in the novel enhancers in LUHMES (Fig. EV2B), and NEUROD1 chromatin immunoprecipitation sequencing (ChIP-seq) data showed higher binding intensity in the novel enhancer regions compared to all previously identified enhancer regions (Fig. EV2C). The *NEUROD1* promoter was highly expressed on Day 1 but showed minimal expression on Day 6 (Fig. EV2A). These findings suggest that NEUROD1 plays a specific role in the early stage of neuronal differentiation. In contrast, upregulated promoters and enhancers shared the enrichment of the same TF binding motifs. Both promoters and enhancers in Cluster 3 were enriched with the binding motifs of Atf and Jun, members of the AP-1 transcriptional complex activated by WNT signaling (Arredondo et al, 2020; Hankey et al, 2018; Liu et al, 2021). The *ATF7* promoter was also classified in Cluster 3, suggesting that it may act as a regulator in the later stage of neuronal differentiation (Fig. EV2A). In the FANTOM5 data, *ATF7* was most highly expressed in the cerebellum (Consortium et al, 2014). Interestingly, the RFX binding motif was enriched in both promoters and enhancers in Cluster 4, suggesting that RFX TFs contribute to activating both promoters and enhancers in the later stages of neuronal differentiation. RFX TFs have well-described functions in neuronal development across various organisms (Choksi et al, 2014; De Stasio et al, 2018; Senti and Swoboda, 2008; Sugiaman-Trapman et al, 2018) and have been implicated in neurodevelopmental and neuropsychiatric disorders (Harris et al, 2021; Tammimies et al, 2016). Furthermore, we recently demonstrated that *RFX2* knockout leads to delayed neuronal differentiation in LUHMES, indicating the crucial role of RFX2 in neuronal development (Coschiera et al, 2024).

To confirm these findings, we evaluated the motif activities of promoters and enhancers using motif activity response analysis (MARA) (Alam et al, 2020; Balwierz et al, 2014; Consortium et al, 2009). By integrating gene expression data with TF binding motif statistics, MARA outputs Z-values that represent the contribution of each motif to gene expression variations across the time course. Here, RFX TFs showed high Z-values in both promoters and enhancers, indicating significant changes in binding motif activity throughout the time course (Fig. 3C). The motif activity of RFX increased during neuronal differentiation, consistent with the above findings (Appendix Fig. S1A) and a previous report (Lauter et al, 2020). Interestingly, PITX TFs showed the highest Z-value in

enhancers, but low in promoters. The motif activity of PITX increased in enhancers but remained unchanged in promoters (Appendix Fig. S1B). Pitx3 is known to be selectively expressed in mouse midbrain dopaminergic neurons and plays a crucial role in their terminal differentiation (Wang et al, 2023). These findings suggest that PITX TFs specifically activate enhancers but not promoters during neuronal differentiation.

## Enhancers associated with GWAS variants for neuronal disorders

It has been shown that disease-associated SNPs identified by GWAS are overrepresented in enhancers active in specific cell types (Ernst et al, 2011; Maurano et al, 2012). Using the Genomic Regulatory Elements and Gwas Overlap algoRithm (GREGOR) tool (Schmidt et al, 2015), we evaluated the enrichment of disease-associated SNPs within (i) enhancers identified in the FANTOM5 project (65,423) (Andersson et al, 2014; Arner et al, 2015), (ii) enhancers newly identified by the NET-CAGE development study (NET_dev; 20,363) (Hirabayashi et al, 2019), and (iii) all enhancers expressed in LUHMES (LUHMES_all; 47,350). Disease-associated SNPs with *P*-values $< 5.0 \times 10^{-8}$ were obtained from the NHGRI-EBI GWAS catalog (downloaded on March 15, 2024) (Buniello et al, 2019). Interestingly, SNPs associated with neuronal disorders, such as Parkinson's disease, schizophrenia, bipolar disorder, and major depressive disorder, were most significantly overrepresented in the enhancers expressed in LUHMES (Fig. 4A). Given that LUHMES is a human midbrain-derived dopaminergic neuronal cell line, it is reasonable to expect that SNPs associated with Parkinson's disease are enriched in the enhancers expressed in LUHMES. It is also known that certain aspects of schizophrenia are caused by dysregulation of the midbrain dopaminergic system (Sonnenschein et al, 2020). Multiple sclerosis also affects the central nervous system, but its associated SNPs were most significantly overrepresented in the enhancers identified in the FANTOM5 project (Fig. 4B). This is likely because multiple sclerosis is an autoimmune disease, and FANTOM5 includes many immune cell samples. SNPs associated with asthma and other autoimmune disorders, such as Crohn's disease and rheumatoid arthritis, also showed the highest enrichment in the FANTOM5 enhancers (Fig. 4B). We further evaluated the enrichment of these disease-associated SNPs within (i) all promoter regions (184,827), (ii) promoter regions expressed in LUHMES (52,076), and (iii) promoter regions differentially expressed in LUHMES (21,907). Except for Parkinson's disease, neuronal disorder-associated SNPs were enriched in promoter regions differentially expressed in LUHMES. Interestingly, autoimmune disease-associated SNPs were not enriched in these differentially expressed promoters (Fig. EV3). Our findings are highly valuable because compared to the number of studies that have examined autoimmune disease-associated SNPs

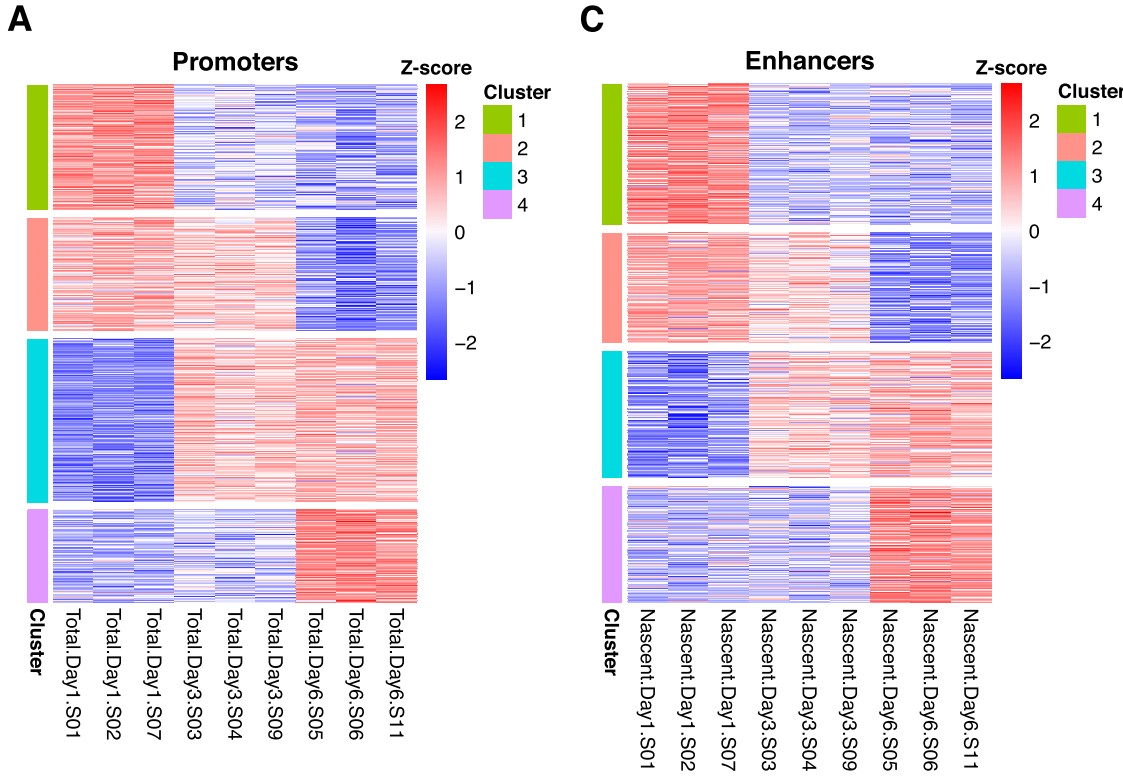

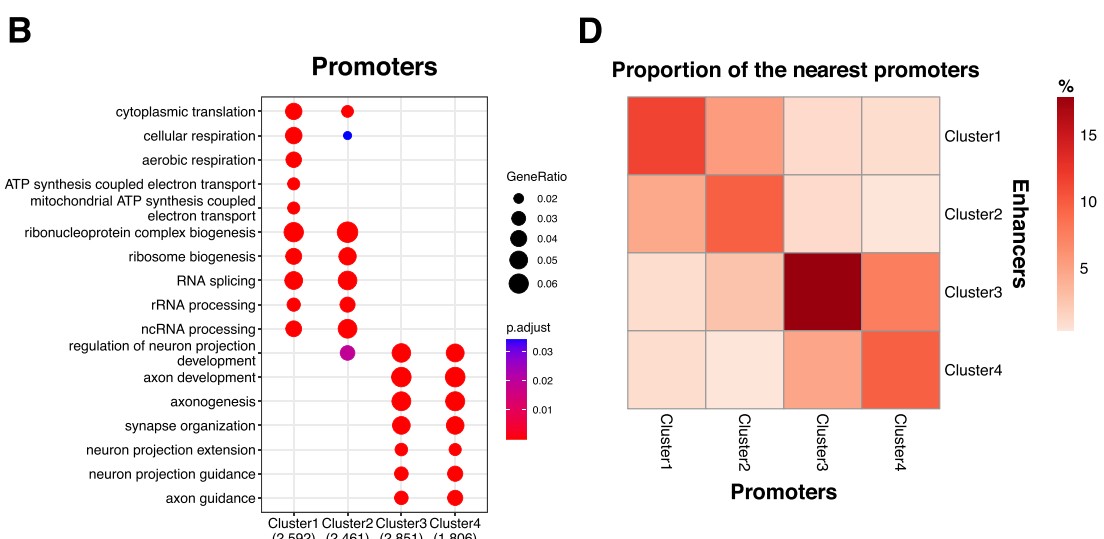

**Figure 2.  Dynamic changes in promoters and enhancers during LUHMES neuronal differentiation.**

(A) Heatmap of differentially expressed promoters during LUHMES differentiation. These promoters were clustered into four groups based on their temporal expression patterns. 'Total' refers to total RNA measured by CAGE. S01–S11 indicate sample numbers. (B) Gene ontology (GO) term enrichment analysis of differentially expressed promoters in each cluster. P-values were calculated using the hypergeometric test and adjusted using the Benjamini-Hochberg method. (C) Heatmap of differentially expressed enhancers during LUHMES differentiation. These enhancers were also clustered into four groups based on their temporal expression patterns. 'Nascent' refers to nascent RNA measured by NET-CAGE. S01–S11 indicate sample numbers. (D) Heatmap illustrating the proportion of overlap between the promoters nearest to enhancers (row) and promoters (column) in each cluster.

in immune cells (Dey et al, 2022; Farh et al, 2015; Nasser et al, 2021; Pelikan et al, 2018), fewer have analyzed neuronal disorder-associated SNPs within enhancers relevant to dopaminergic neurons (Dong et al, 2018; Girdhar et al, 2022).

We further sought to characterize the 31,057 novel putative enhancer regions. First, we found these putative enhancer regions were significantly more conserved compared to randomly selected coding and genome-wide regions (Fig. EV4A). Next, we observed a

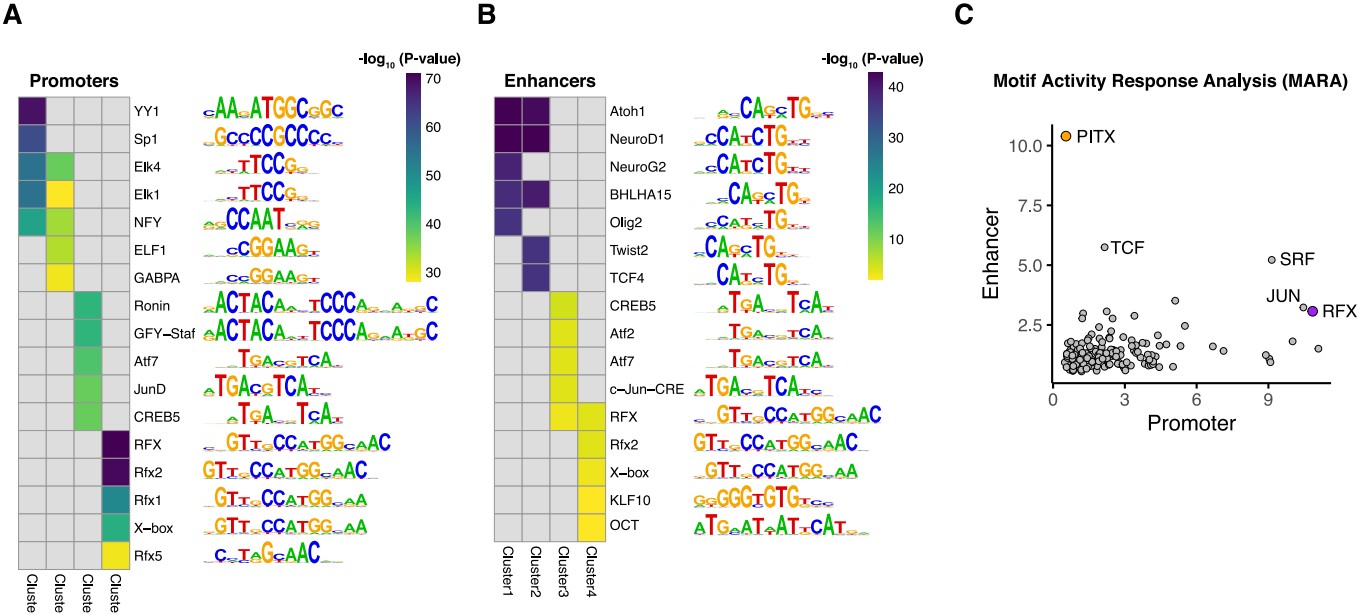

**Figure 3. Transcription factor binding motif analysis of promoters and enhancers during LUHMES neuronal differentiation.**

(A, B) Heatmap showing the enrichment of transcription factor binding motifs in differentially expressed promoters (A) and enhancers (B) during LUHMES differentiation. The top five significantly enriched motifs in each cluster are shown. *P*-values were calculated using the binomial test. (C) Z-values indicating transcription factor binding motif activity changes in promoters (x-axis) and enhancers (y-axis) during LUHMES differentiation.

higher number of SNPs within these putative enhancer regions than in the two random sets, suggesting considerable genetic variability. However, when evaluating nucleotide diversity for SNPs within these regions, we found that the putative enhancer regions displayed significantly lower diversity than the random sets (Fig. EV4B). This finding suggests that the putative enhancer regions may be relatively intolerant to variation and likely subject to selective pressure. Furthermore, we found these SNPs were significantly enriched with substantia nigra-specific expression quantitative trait loci (eQTLs) compared to the random sets (Fig. EV4C). Taken together, these results support the idea that these putative enhancers may play functionally significant roles, particularly in midbrain dopaminergic neurons.

## Identification of enhancer–promoter pairs by Capture Hi-C

We used Capture Hi-C to identify target genes of enhancers during LUHMES neuronal differentiation. Using a capture probe set, we targeted 19,495 promoters, 6019 enhancers, and 3000 negative controls (regions with no annotated regulatory activity) and performed Capture Hi-C at three time points (Days 1, 3, and 6) in two replicates. The targeted 6018 enhancers were either (i) differentially expressed (2172), (ii) overlapping with H3K27ac peaks in differentiated LUHMES (1629) (Pierce et al, 2018), or (iii) overlapping with GWAS variants associated with neuronal disorders such as Parkinson's disease and schizophrenia (2218). We used HiCapTools to detect significant interactions (Anil et al, 2018). HiCapTools derives a background interaction frequency distribution using the interactions of negative control regions, instead of a theoretical background interaction frequency used in

similar software packages such as CHiCAGO (Freire-Pritchett et al, 2021). In our benchmark study, we show that HiCapTools performs similarly to CHiCAGO in detecting short and mid-range enhancers but performs better at detecting distal enhancers (distance span >500 kb). HiCapTools does not require multiple-testing correction, which also increases accuracy (Anil et al, 2018). Using HiCapTools, we detected 68,406 interactions (minimum supporting pair = 5 and Bonferroni $P = 0.1$, corresponding to approximately 3.7-fold enrichment with respect to negative controls). The breakdown of interactions per target can be found in Fig. 5A. The average interaction distance is 99.8 kb, with 32% of the interactions spanning longer than 100 kb (Fig. 5B). Only 24.1% (9901 out of 40,972) of the promoter interactions involved the nearest genes. Except for cluster 2, enhancers most frequently interacted with promoters belonging to the same cluster (Fig. EV5A). Enhancers in cluster 2 interacted most frequently with promoters in cluster 1, both of which showed a downregulation pattern. We overlapped the promoter-interacting regions (excluding targeted regions) with neural enhancer datasets from ChIP-Atlas (Zou et al, 2024) (see Methods) (Fig. 5C). Promoter-interacting regions were 7.4-fold and 5.2-fold enriched for composite peaks (i.e., peaks overlapping all type of enhancer peaks) and LUHMES putative enhancers, respectively, amounting to 79.8% of the promoter-interacting regions overlapping at least one of the enhancer peaks, indicating accurate capturing promoter–enhancer interactions.

We observed that 36% (24,908 out of 68,406) of the interactions involved either a differentially expressed promoter or putative enhancer in LUHMES cells, suggesting a significant role for chromatin interactions during differentiation. To quantify the degree of interaction changes across the time points, we calculated the Jaccard index (JI) and overlap coefficient (OCE), which are commonly used to

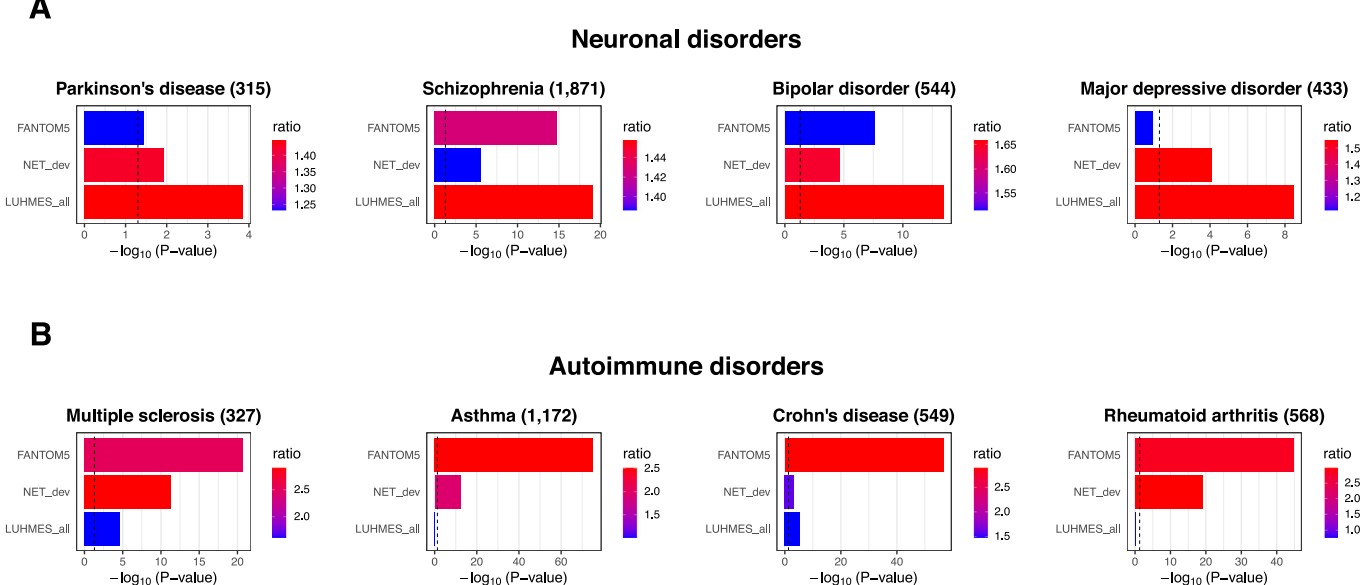

**Figure 4. Enrichment of neuronal disorder-associated GWAS SNPs within enhancers active in LUHMES.**

Enrichment of GWAS SNPs associated with neuronal (A) or autoimmune disorders (B) in the enhancer regions identified by the FANTOM5 project (65,423), the NET-CAGE development paper (NET_dev; 20,363), and all expressed enhancers in LUHMES (LUHMES_all; 47,350). P-values were calculated using the permutation test. The dashed lines represent P = 0.05. Numbers in parentheses indicate the number of SNPs for each disorder.

quantify the similarities between sets (Gupta and Sardana, 2015; Zhigulev et al, 2024) (see Methods). For each interacting feature, we calculated the abovementioned similarity indexes across three time points (Day 3 vs Day 1, Day 6 vs Day 1, and Day 6 vs Day 3) and asked if promoters of differentially expressed genes are more likely to change their interactions. Indeed, the interactome of differentially expressed genes was significantly different across time points compared to that of non-differentially expressed genes (Fig. 5D). For example, *RET* gene is a tyrosine kinase receptor for glial cell line-derived neurotrophic factors (GDNFs), which are essential for the development and maintenance of midbrain dopaminergic neurons (Drinkut et al, 2016; Li et al, 2006; Lin et al, 1993). *RET* gene was differentially expressed at all time points in LUHMES (Dataset EV4), and accordingly, its interaction profile differed between the time points (Fig. 5E). On Day 1, *RET* engaged in long-range interactions (thirteen contacts spanning 102–230 kb distance), along with four short-range interactions that were also present on Day 6. On Day 3, the *RET* promoter had only one contact (135 kb away), losing the rest of both short- and long-range contacts. On Day 6, *RET* promoters regained only the four short-range interactions that were present on Day 1.

We then focused on the interaction profile of targeted LUHMES putative enhancers (denoted as "Enhancer" and "DE Enhancer" in Fig. 5A). In total, they had 11,656 interactions connecting to 2254 LUHMES enhancers, 1769 promoters, and 180 GWAS variants. Of the differentially expressed enhancers, 43.9% (955 out of 2172) were involved in an interaction. Genes interacting with differentially expressed enhancers were more likely to be differentially expressed (25.6% vs. 37.7%, respectively; $P = 2.7 \times 10^{-46}$, Dataset EV4), and the expression levels of interacting promoters and enhancers in a dynamic manner (i.e., the interaction was observed only at a single time point), showed the strongest positive correlation at the time point when the interaction between them

was observed (Fig. 6A). Notably, 1243 genes interacted with the enhancers and were significantly enriched for genes associated with neuronal differentiation, including axon development and axonogenesis (Fig. EV5B).

We next focused on interactions of GWAS variants associated with neurodevelopmental disorders. We targeted 2218 variants or their linkage disequilibrium counterparts (Dataset EV5) using sequence capture probes; 49% (1082) of them yielded at least one interaction. To determine if the interactions of GWAS variants changed during differentiation at different time points, we compared the gene enrichment categories of their target genes between different time points. Indeed, GWAS-target gene enrichments differed between time points (Fig. 6B). For example, terms related to synaptic signaling were only enriched for genes connected to GWAS variants on Day 6. The GWAS variants represented a complex interaction profile, as exemplified by one schizophrenia-associated GWAS variant (rs1261117) that interacted with the promoter of the differentially expressed TF gene *TCF4* (transcription factor 4) at all time points (Fig. 6C). However, on Day 3 only, the SNP exhibited a long-range interaction (1.2 Mb) with a differentially expressed enhancer (chr18:54230801–54231236). In human induced pluripotent stem cells (ENCODE), the distal enhancer is bound to chromatin modifiers such as HDAC2 and KDM1A, cohesin components such as CTCF and RAD21, and specific TFs such as SIX5 and MYC (Wang et al, 2012). The *TCF4* gene encodes a TF important for neural differentiation (Mesman et al, 2020). Its haploinsufficiency causes Pitt–Hopkins syndrome, a disease characterized by intellectual disability, developmental impairment, recurrent seizures, and breathing abnormalities (Zweier et al, 2007). Various exonic variants of the gene have already been associated with schizophrenia and bipolar disorder (Quednow et al, 2014).

*SNCA* (synuclein alpha) is known as the major causative gene of Parkinson's disease (Siddiqui et al, 2016). In LUHMES, seven

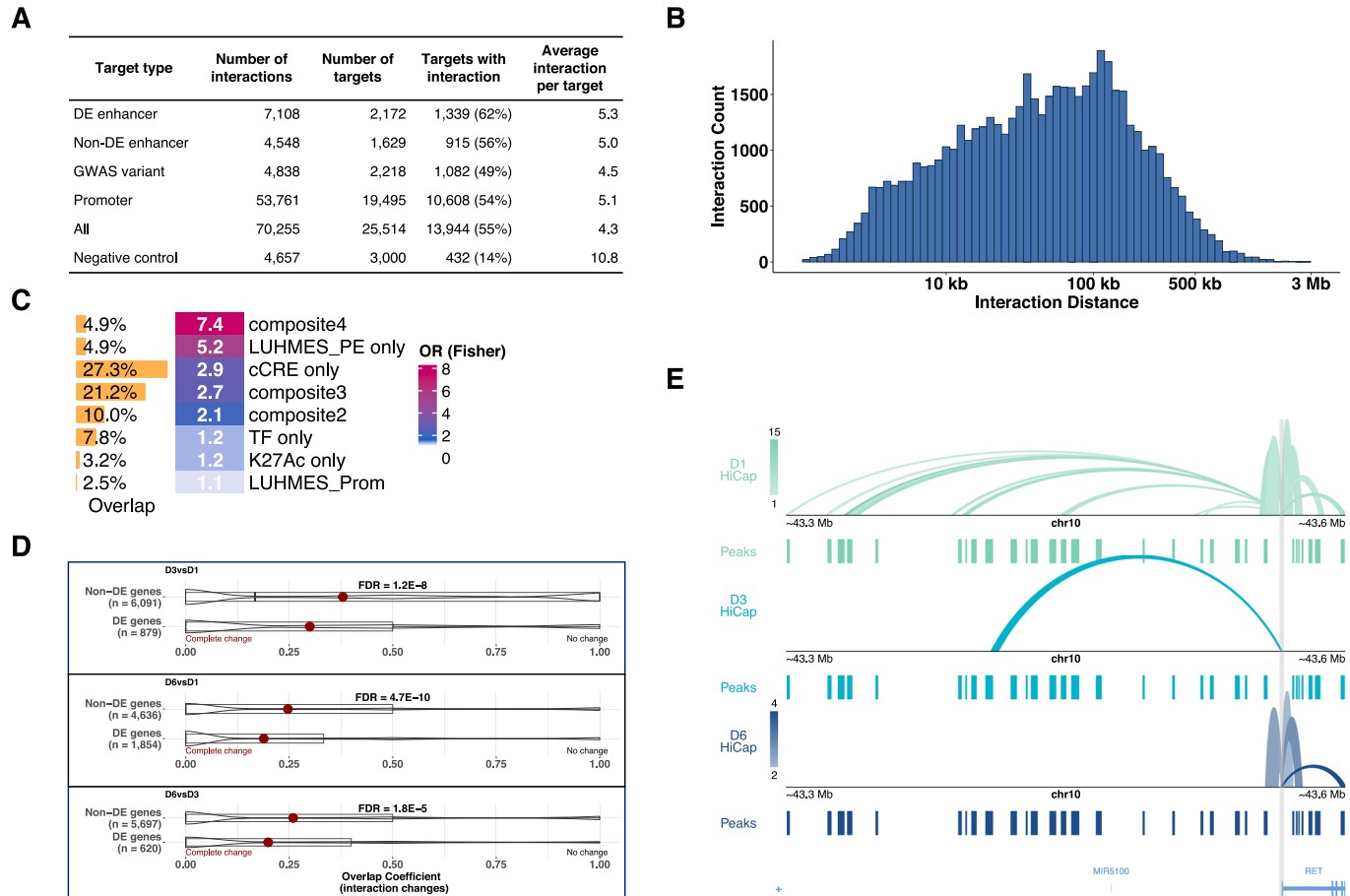

Figure 5. The genome interaction landscape of enhancers in LUHMES neuronal cells is significantly altered upon differentiation.

(A) The breakdown of targeted features and the number of interactions for each feature type. DE Enhancer: differentially expressed enhancer. (B) The distribution of interaction distances. Absolute values of the interaction distances are taken, and logarithmic values of the distances are plotted using a bin size of 500 bases. (C) The functional enrichment profile of the distal regions. The first column shows the percentage of overlap between the functional set and the distal interactors. The heatmap on the right shows the fold-enrichment of the distal interactors for the functional set. The full results are found in Dataset EV10. Composite2, composite3, and composite4 correspond to regions where any two, three, or all four functional marks overlap, respectively (except promoters). LUHMES_PE: LUHMES putative enhancers, cCRE: cis-regulatory elements (ENCODE), TF: transcription factor ChIP peaks (Neural), K27Ac: H3K27Ac ChIP peaks, LUHMES_Prom: LUHMES promoters. (D) The extent of interactome changes between differentially expressed (DE) and non-DE promoters across time points. DE genes were more likely to change their interactions compared to non-DE genes, as measured by the overlap coefficient. (E) The interactions of the RET gene across time points. Peaks correspond to those used in the enrichment analysis. The full list can be found in Dataset EV10.

alternative promoters were differentially expressed during differentiation (Appendix Fig. S2). We found that the p7/p13 promoters significantly interacted with a novel enhancer active in LUHMES (chr4:90721375–90721795). Notably, rs2583959 and rs2737024, both of which associated with Parkinson's disease, are located within this enhancer region (McClymont et al, 2018). Furthermore, this enhancer region overlaps with a mouse midbrain-specific enhancer region (mm9: chr6: 60742503–60744726) that was suggested to interact with the *Snca* promoter (McClymont et al, 2018).

*POU6F2* (POU class 6 homeobox 2) is known to be expressed in a limited number of cell types (Yoshihara et al, 2017) but is also reportedly expressed in the developing midbrain (Zhou et al, 1996). We found that rs7786896, located within the intronic region of *POU6F2* and associated with schizophrenia (Lu et al, 2020), interacts with the alternative promoters (p1/p2/p4) of the *POU6F2*

gene (Appendix Fig. S3). Although the interaction was weaker on Day 1 (supporting pair = 4), this variant also overlapped with the bidirectionally transcribed locus chr7:3909354–39094330, which may act as an enhancer on Day 1.

In addition, we focused on two genes, *ZMIZ1* (zinc finger MIZ-type containing 1) and *MAP7D1* (MAP7 domain containing 1). ZMIZ1 protein regulates the activity of various TFs, including the androgen receptor and p53. It was recently reported that mutations in *ZMIZ1* can cause a rare neurodevelopmental syndrome characterized by growth failure, feeding difficulties, microcephaly, facial dysmorphism, and various other congenital malformations (Carapito et al, 2019). Our Capture Hi-C data demonstrated that the alternative promoters (p7/p9) of the *ZMIZ1* gene interact with the FANTOM5-defined enhancer chr10:80997251–80997825 at all time points (Appendix Fig. S4). *MAP7D1*, predicted to be involved in microtubule cytoskeleton organization, was recently identified to

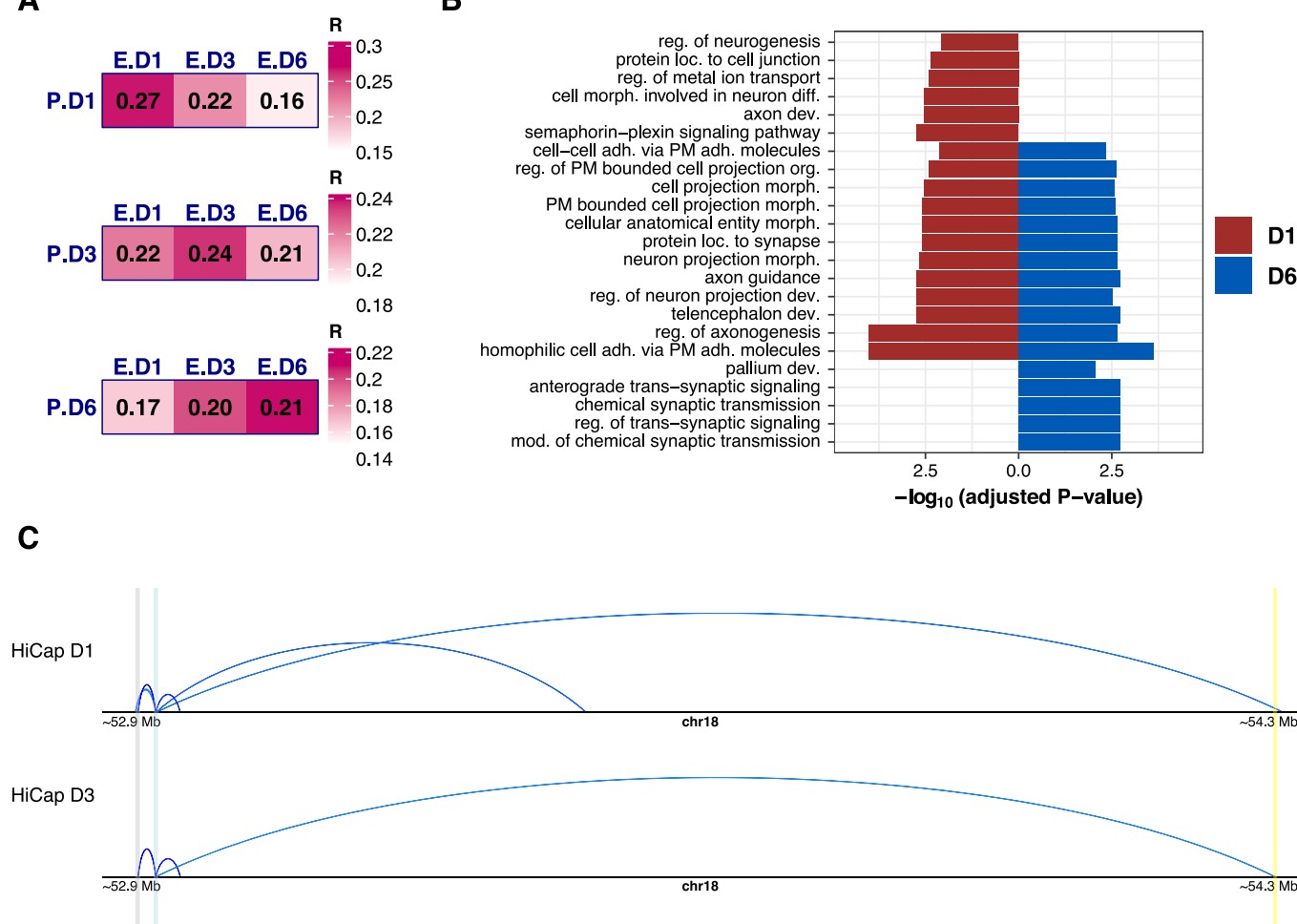

**Figure 6. Differentially expressed GWAS enhancers alter their connectivity during LUHMES neuronal differentiation.**

(A) Expression levels of interacting features were most strongly correlated at the time point when the interaction was present. P.D1, P.D3, P.D6: promoter expression at Day 1, Day 3, Day 6; E.D1, E.D3, E.D6: NET-CAGE peak expression at Day 1, Day 3, Day 6. (B) Gene ontology (GO) term enrichment of genes interacting with GWAS variants on Day 1 and Day 6 was analyzed separately to assess changes in their target genes. P-values were calculated using the hypergeometric test and adjusted using the Benjamini-Hochberg method. GWAS target genes showed specific and relevant enrichments at different time points. (C) Interaction profile of a differentially expressed *TCF4* promoter (highlighted in blue) involving dynamic interactions with a differentially expressed enhancer located 1.2 Mb away (highlighted in yellow) and a GWAS variant associated with schizophrenia (rs1261117, highlighted in gray). Only differentially expressed (DE) promoters and enhancers are displayed in the DE promoter and the DE enhancer tracks. Note that the long-range interaction on Day 1 involves a separate region (~9.3 kb away).

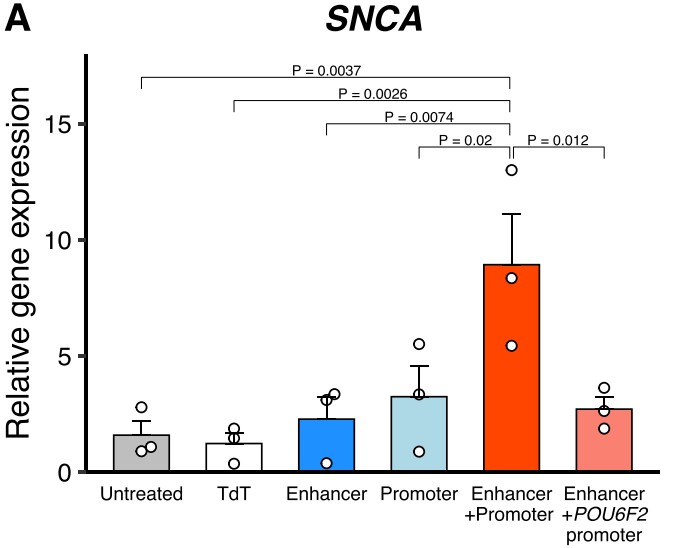

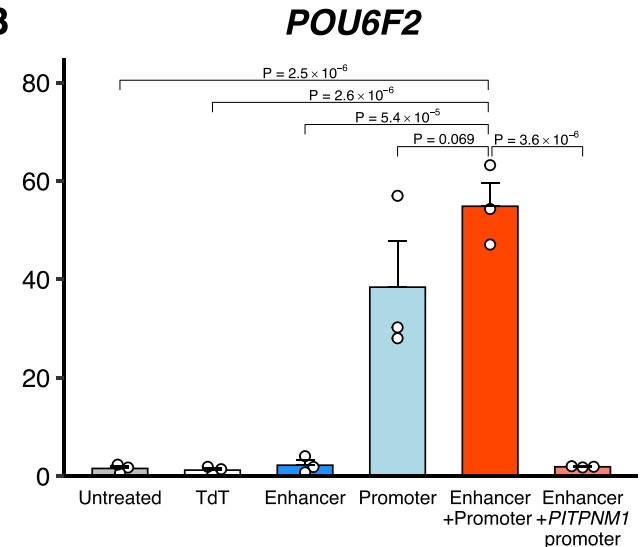

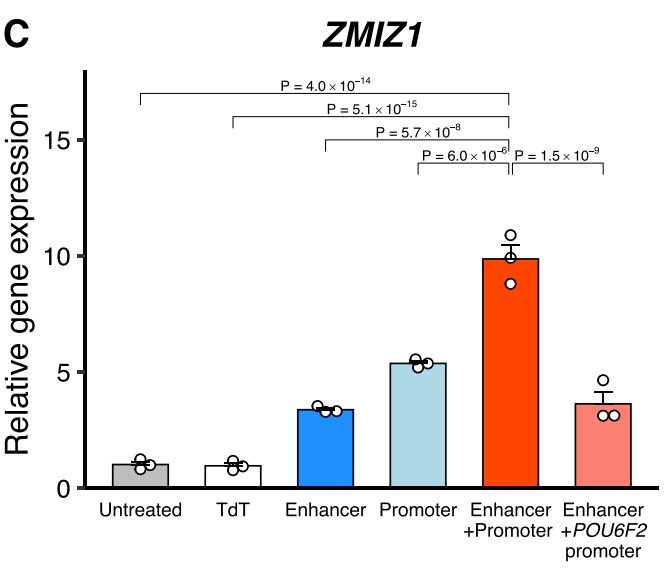

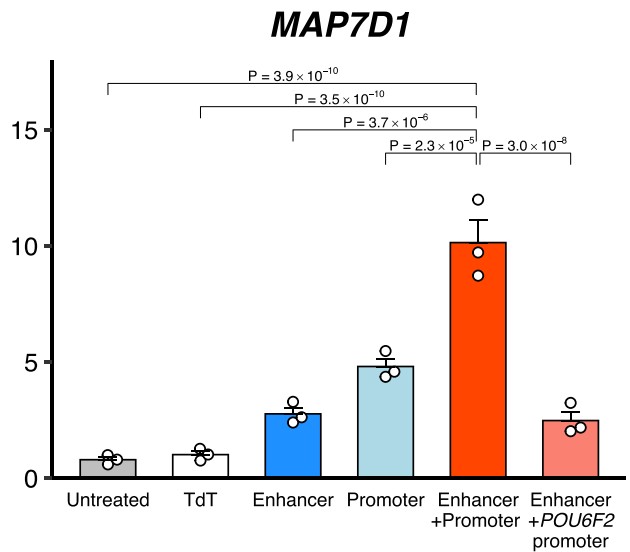

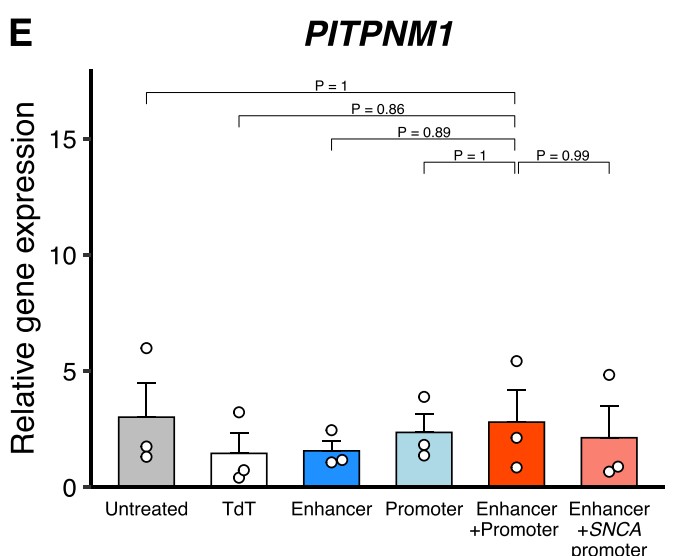

◄ **Figure 7. Enhancer–promoter interactions validated by CRISPRa.**

Regulation of mRNA levels of the *SNCA* (**A**), *POU6F2* (**B**), *ZMIZ1* (**C**), *MAP7D1* (**D**), and *PITPNM1* (**E**) genes measured by qRT-PCR in hTERT-RPE1 cells 24 h after transfection with gRNAs and the dCas9VP192 plasmid. Error bars represent the standard error of the mean, $n = 3$ independent experiments. Statistical significance was assessed using Dunnett's multiple comparisons test following a one-way ANOVA. Source data are available online for this figure.

be associated with schizophrenia by a transcriptome-wide association study (Gusev et al, 2018). Again, our Capture Hi-C data demonstrated that a promoter (p9) of the *MAP7D1* gene interacts with the FANTOM5-defined enhancer chr1:36627717–36628369 on Day 1 (Appendix Fig. S5). These observations prompted us to verify not only whether these regions act as enhancers but also whether these enhancer–promoter pairs actually interact.

## Validation of functional enhancers

To confirm the sets of functional enhancer–promoter pairs, including several enhancer-associated SNPs implicated in neuronal disorders, we tested the function of enhancers in vitro using the CRISPR activation (CRISPRa) method (Weltner et al, 2018). We used human TERT (hTERT)-RPE1 cells because they are easy to transfect with exceptionally high transformation rates, whereas LUHMES cells are comparatively difficult to transfect (Lauter et al, 2020; Shah et al, 2016). Both cell lines are non-malignant with a normal karyotype, originate from the neuroectoderm, and exhibit very similar characteristics with respect to proliferation and differentiation into the respective mature cell types. The dCas9-VP192 activator (Balboa et al, 2015; Weltner et al, 2018) was used to transfect cells in combination with a pool of four specific guide RNAs (gRNAs) targeting the selected enhancer as well as the proximal promoter region of the *SNCA*, *POU6F2*, *ZMIZ1*, and *MAP7D1* genes (Appendix Figs. S2–5; Dataset EV6). We also designed specific gRNAs targeting an unreplicated interaction of the *PITPNM1* gene as a negative control. Gene expression was assayed by quantitative reverse transcription PCR (qRT-PCR) 24 h after transfection.

Activation of the *SNCA* enhancer in combination with the *SNCA* promoter led to significant upregulation of *SNCA* expression compared both to the control (TdTomato: TdT) ($P = 0.0026$, Dunnett's multiple comparisons test) and cells transfected with enhancer gRNAs alone ($P = 0.0074$) (Fig. 7A). Similarly, we observed significant increases in *POU6F2*, *ZMIZ1*, and *MAP7D1* gene expression induced by activation of the enhancer in combination with the promoter, compared to both the control and cells transfected with enhancer gRNAs alone ($P < 0.001$) (Fig. 7B–D). This increase was also observed compared to cells with the enhancer activated in combination with the promoter of a non-specific gene. As expected, no significant change in *PITPNM1* gene expression was observed in cells transfected with *PITPNM1* enhancer gRNAs (chr11:67292367–67293218) in combination with specific promoter gRNAs (Fig. 7E). These findings indicated that these variants are located within truly functional enhancers that regulate the expression of *SNCA*, *POU6F2*, *ZMIZ1*, and *MAP7D1* in differentiating neurons.

## Discussion

The characterization of enhancers as gene-regulating elements has revealed that there are many more enhancers than protein-coding

genes or promoters, and that enhancers function in a highly cell type-specific manner. These observations help explain the large number of different cell types in humans and the complexity of differentiation programs. Using the FANTOM5 dataset, brain region-specific enhancers have been identified, and these regions were enriched with variants associated with autism spectrum disorders (Yao et al, 2015). The surprisingly high rate of GWAS variants coinciding with enhancers rather than coding regions has spurred great interest in better understanding gene regulation by enhancers, and new enhancers continue to be identified. Therefore, we sought to use a well-characterized, non-malignant human neuronal differentiation model to explore the enhancers driving this well-defined process. Surprisingly, our results identified 31,057 previously unannotated putative enhancers, nearly tripling the number of known active enhancers by applying the NET-CAGE approach to the time course of LUHMES human dopaminergic cell differentiation. Moreover, our results indicated a significant overlap between the enhancers active in human LUHMES neuronal cells and GWAS variants, particularly those associated with Parkinson's disease and schizophrenia.

GWAS-based identification of large numbers of susceptibility alleles for numerous complex disorders has revealed the complex genetic architecture of these disorders and also highlighted the role of non-coding regulatory regions in their pathogenesis. However, such potential regulatory associations provide few insights into the pathogenesis of these disorders unless the active elements are identified and their corresponding regulated genes are linked to their regulatory regions.

Here, we demonstrate that a differentiating human neuronal cell model can be used to identify a large set of novel neuronal enhancers that overlap with GWAS variants, particularly those related to disorders of the dopaminergic system (Beaulieu and Gainetdinov, 2011; Perez de la Mora et al, 2020). Although the enhancers themselves may not serve as potential drug targets, the genes they regulate might; thus, linking GWAS variants to genes through enhancers could provide new druggable targets. The regulatory links reported here represent only the tip of the iceberg in terms of aiming to identify new drug targets for treating these devastating disorders. A clear limitation of this study is its scope, which focuses on only one, albeit very well-defined and described, human neuronal differentiation model. As many of the enhancers identified here may have more general neuronal roles, we cannot suggest a direct link between dopaminergic regulation and any of the disease associations. We anticipate that research on other neuronal cell models or human brain tissue samples will uncover a large number of yet-unidentified enhancers with key roles in differentiation, regulation of different brain regions, and disease pathogenesis.

Our study included three main phases: (i) enhancer identification, (ii) enhancer–promoter linking, and (iii) functional validation of select enhancer–promoter pairs for gene regulation. To identify putative enhancers, we used the NET-CAGE protocol, which has a high sensitivity in detecting eRNAs. A number of technologies have been developed to identify putative enhancers, each with its own

advantages and limitations. For instance, putative enhancers identified by chromatin accessibility or TF binding can include various regulatory elements in addition to enhancers, while those identified by histone modifications are often broader, making it difficult to detect enhancers with high nucleotide resolution (reviewed in (Murakawa et al, 2016)). Although NET-CAGE can identify only transcribed enhancers, these enhancers are more likely to be active (Andersson et al, 2014). Nevertheless, false positives may arise due to transcriptional noise or bidirectional transcription from insulators (Melgar et al, 2011) and accessible DNA (Young et al, 2017), while false negatives may result from eRNAs expressed at very low levels below detection thresholds or from enhancers that do not produce eRNAs (Catarino and Stark, 2018). For enhancer–gene linking, we used a modified Capture Hi-C method with much higher promoter/enhancer resolution, but it required targeting the 19,495 promoters and 6019 selected enhancer elements based on their overlap with GWAS variants or their differential expression status. It should be noted that spatial proximity does not always reflect functional regulatory relationships. Therefore, to functionally validate the enhancer effects on transcription, we employed a CRISPRa-based methodology using gRNA molecules targeting either putative enhancers, promoters, or both. Including gRNAs for both a putative enhancer and its coupled promoter in a CRISPRa experiment resulted in significantly higher transcription than using either gRNA alone.

Our study further exemplifies the power of enhancer discovery in providing potential clues to better understand the pathogenesis of neuropsychiatric diseases. This study reveals fine-tuned regulation of neuronal differentiation even between adjacent developmental stages, occurring just a few days apart. Taken together, these results emphasize the vast regulatory potential embedded in noncoding DNA which houses relevant enhancers, and provide a rich resource for further studies on neuronal development, regulation, and disorders.

# Methods

### Reagents and tools table

| Reagent/Resource | Reference or Source | Identifier or Catalog Number |
| --- | --- | --- |
| **Experimental models** | | |
| LUHMES cells (*H. sapiens*) | ATCC | CRL-2927 |
| hTERT RPE-1 cells (*H. sapiens*) | ATCC | CRL-4000 |
| **Recombinant DNA** | | |
| CAG-dCasVPN192-T2A-EGFP | Weltner and Trokovic (2021) | |
| **Oligonucleotides and other sequence-based reagents** | | |
| Hi-C probes | This study | Dataset EV5 |
| SureSelect Custom Tier3 Probes | Agilent | 5191-6919 |
| gRNA oligos | This study | Dataset EV6 |
| qPCR primers | This study | Dataset EV9 |
| **Chemicals, Enzymes and other reagents** | | |
| poly-L-ornithine hydrobromide | Sigma-Aldrich | P3655 |

| Reagent/Resource | Reference or Source | Identifier or Catalog Number |
| --- | --- | --- |
| human fibronectin | Sigma-Aldrich | F1056 |
| DMEM/F-12 Ham growth medium | Sigma-Aldrich | D6421 |
| L-glutamine | Sigma-Aldrich | G7513 |
| N-2 supplement | Gibco | 17502-048 |
| human basic fibroblast growth factor (bFGF) | Thermo Fisher Scientific | PHG0369 |
| tetracycline hydrochloride | Sigma-Aldrich | T7660 |
| hygromycin B | Gibco | 10687010 |
| TrypLE Express enzyme | Thermo Fisher Scientific | 12605010 |
| Nuclei EZ Lysis Buffer | Sigma-Aldrich | NUC101 |
| α-amanitin | FUJIFILM Wako Pure Chemical | 010-22961 |
| cOmplete Protease Inhibitor Cocktail | Roche | 4693116001 |
| cOmplete EDTA-free Protease Inhibitor Cocktail | Roche | 4693159001 |
| SUPERaseIN RNase Inhibitor | Invitrogen | AM2694 |
| QIAzol lysis reagent | Qiagen | 79306 |
| Empigen BB detergent | Sigma-Aldrich | 30326 |
| DNase I | Invitrogen | AM2222 |
| formaldehyde | Thermo Fisher Scientific | 28906 |
| FastDigest MboI | Thermo Fisher Scientific | FD0814 |
| Klenow enzyme | Thermo Fisher Scientific | EP0051 |
| biotin-14-dATP | Thermo Fisher Scientific | 19524-016 |
| T4 DNA ligase | BioNordika | M0202L |
| Phenol:Chloroform:Isoamyl alcohol sol | Sigma-Aldrich | 77617-100ML |
| RNAse A | Thermo Fisher Scientific | EN0531 |
| Ampure XP beads | Beckman Coulter | A63881 |
| Dynabeads MyOne Streptavidin C1 beads | Thermo Fisher Scientific | 65001 |
| Phusion Polymerase | Thermo Fisher Scientific | F350 |
| Lipofectamine 3000 transfection reagent | Invitrogen | L3000001 |
| Opti-MEM medium | Gibco | 31985062 |
| **Software** | | |
| Cutadapt (v2.3) | https://cutadapt.readthedocs.org/ Martin (2011) | |
| rRNAdust script (v1.06) | https://fantom.gsc.riken.jp/5/suppl/rRNAdust/rRNAdust1.06.tgz | |
| STAR (v2.5.1b) | Dobin et al (2013) | |
| bigWigAverageOverBed | Kent et al (2010) | |
| edgeR (v3.40.2) | Robinson et al (2010) | |

| Reagent/Resource | Reference or Source | Identifier or Catalog Number |
|---|---|---|
| clusterProfiler (v4.7.1.3) | Wu et al (2021) | |
| HOMER (v4.11) | Heinz et al (2010) | |
| MARA | Alam et al (2020) | |
| deepTools (v3.5.2) | Ramírez et al (2016) | |
| GREGOR (v1.4.0) | Schmidt et al (2015) | |
| BEDtools (v2.31.1) | Quinlan and Hall (2010) | |
| bcftools (v1.21) | Danecek et al (2021) | |
| bowtie2 (v2.2.9) | Langmead et al (2009) | |
| HiCUP | Anil et al (2018) | |
| plotgardner (v1.8.2) | Kramer et al (2022) | |
| valr (v0.8.1) | Riemondy et al (2017) | |
| igraph (v1.4.2) | Csárdi and Nepusz (2006) | |
| GenomicFeatures (v1.57.0) | Lawrence et al (2013) | |
| RMariaDB (v1.3.2) | https://cran.r-project.org/package=RMariaDB | |
| ChIPpeakanno (v3.39.2) | Zhu et al (2010) | |
| Benchling CRISPR tool | https://benchling.com/ | |
| Graph Pad Prism 10 | https://www.graphpad.com/ | |
| **Other** | | |
| miRNeasy Mini Kit | Qiagen | 217004 |
| RNeasy Mini Kit | Qiagen | 74104 |
| Agilent 2100 Bioanalyzer instrument | Agilent | G2939BA |
| DNA 12000 kit | Agilent | 5067-1508 |
| TruSeq Stranded Total RNA Library Prep Kit with Ribo-Zero Human/Mouse/Rat Set B | Illumina | 20020612 |
| Qubit DNA high sensitivity kit | Thermo Fisher Scientific | Q32851 |
| SureSelect XT HS2 DNA Kits for Library Preparation and Target Enrichment | Agilent | G9982A |
| Illumina NextSeq 500 | Illumina | 20024906 |
| Illumina NovaSeq 6000 | Illumina | 151-10-10-151 |
| NanoDrop 8000 Spectrophotometer | Thermo Fisher Scientific | ND-8000-GL |
| RevertAid H Minus First Strand cDNA Synthesis Kit | Thermo Fisher Scientific | K1631 |
| 7500 Fast Real-Time PCR System | Applied Biosystems | |

## Cell cultures

The LUHMES neuronal cell line was obtained from the American Type Culture Collection (ATCC; CRL-2927, RRID: CVCL_B056) and cultured in standard conditions (37 °C, 5% $CO_2$) as previously described (Coschiera et al, 2023; Lauter et al, 2020). LUHMES cells were seeded in flasks pre-coated with poly-L-ornithine hydrobromide (Sigma-Aldrich P3655; 50 µg/mL) and human fibronectin (Sigma-Aldrich F1056; 1 µg/mL), and grown in DMEM/F-12 Ham

growth medium (Sigma-Aldrich D6421) supplemented with L-glutamine (Sigma-Aldrich G7513; 2.5 mM), N-2 supplement (Gibco 17502-048; 1x), and human basic fibroblast growth factor (bFGF) (Thermo Fisher Scientific PHG0369; 40 ng/mL). To differentiate LUHMES neuronal precursor cells into post-mitotic neurons, tetracycline hydrochloride (Sigma-Aldrich T7660; 1 µg/mL) was added to the culture medium instead of bFGF to deactivate and silence v-myc transgene expression.

Human hTERT-immortalized retinal pigment epithelial (RPE1) cells were obtained from ATCC (CRL-4000, RRID: CVCL_4388) and cultured in standard conditions (37 °C, 5% $CO_2$) using DMEM/F-12 Ham growth medium supplemented with 10% fetal bovine serum and hygromycin B (Gibco 10687010; 20 µg/mL) for transgenic selection. All cell lines were confirmed to be free of mycoplasma contamination.

## CAGE and NET-CAGE sample preparation

LUHMES samples for CAGE and NET-CAGE methods were prepared following previously established protocols (Hirabayashi et al, 2019). About $1.5 \times 10^7$ cells/sample were collected at Day 1, Day 3, and Day 6 of differentiation, after 24, 72, and 144 h of tetracycline exposure, respectively. Prior to collection, cells were washed twice with phosphate-buffered saline (PBS), detached using TrypLE Express enzyme (Thermo Fisher Scientific 12605010), and centrifuged at $200 \times g$ for 5 min to remove the supernatant, and then the pellets were flash frozen.

To lyse the cell pellets, 1500 µL of Cell Lysis Buffer composed of Nuclei EZ Lysis Buffer (Sigma-Aldrich NUC101) supplemented with α-amanitin (FUJIFILM Wako Pure Chemical 010-22961; 25 µM), cOmplete Protease Inhibitor Cocktail (Roche 4693116001; 1x), SUPERaseIN RNase Inhibitor (Invitrogen AM2694; 20 U), and sodium lauryl sulfate (0.1%) was used. A 100-µL aliquot was taken and mixed with 700 µL of QIAzol lysis reagent (Qiagen 79306) for subsequent total RNA extraction for CAGE sequencing. The remaining cell lysate was kept on ice for 10 min and centrifuged at $2000 \times g$ for 5 min at 4 °C to remove the supernatant. The pellet was washed with 600 µL of Cell Lysis Buffer and centrifuged again at $2000 \times g$ for 5 min at 4 °C to completely remove the cytoplasmic RNA from the supernatant.

The pellet was then resuspended in 200 µl of Nucleic Lysis Buffer in RNase-free water (1% NP-40, 20 mM HEPES, 300 mM NaCl, 2 M urea, 0.2 mM EDTA, 1 mM DTT) supplemented with α-amanitin (25 µM), cOmplete Protease Inhibitor (1x), SUPERaseIN (20 U), and Empigen BB detergent (Sigma-Aldrich 30326; 0.1%), set on ice for 10 min, and centrifuged at $3000 \times g$ for 4 min at 4 °C to remove the supernatant containing the nucleoplasmic fraction. At this point the pellet represented the chromatin fraction only, and it was washed with 100 µL Nuclei Lysis Buffer and centrifuged at $3000 \times g$ for 4 min at 4 °C to further remove nucleoplasmic RNA.

The pellet was treated with 50 µL of DNase I (Invitrogen AM2222) in RNase-free water solution (1x DNase I buffer, 50 U DNase enzyme) supplemented with SUPERaseIN (20 U) for 30 min at 37 °C, pipetting every 5 min to dissolve the pellet. To isolate nascent-RNAs for NET-CAGE sequencing, 700 µL of QIAzol was added to the solution.

Both nascent and total RNAs were first extracted through phenol–chloroform phase separation, and then the miRNeasy Mini Kit (Qiagen 217004) was used, including the optional on-column

DNase digestion step (Qiagen 79254). RNA integrity was assessed using the 2100 Bioanalyzer instrument (Agilent G2939BA).

## CAGE and NET-CAGE library preparation and sequencing

Libraries were prepared and sequenced according to (Hirabayashi et al, 2019). CAGE complementary DNA (cDNA) libraries were constructed following the no-amplification non-tagging CAGE libraries for the Illumina next-generation sequencers (nAnT-iCAGE) protocol (Murata et al, 2014), while NET-CAGE cDNA libraries were generated using the TruSeq Stranded Total RNA Library Prep Kit with Ribo-Zero Human/Mouse/Rat Set B (Illumina 20020612). Finally, CAGE and NET-CAGE cDNA libraries were sequenced on the NextSeq 500 system with single-end 75-bp length (Illumina 20024906).

## CAGE and NET-CAGE data processing

To reduce the frequency of low-quality reads, Cutadapt (v2.3) (Martin, 2011) (https://cutadapt.readthedocs.org/) was used with the following parameters: --cut -6 --nextseq-trim=20 -m 35. Reads aligned to human ribosomal RNA sequences (GenBank U13369.1) were removed using the rRNAdust script (v1.06) (https://fantom.gsc.riken.jp/5/suppl/rRNAdust/rRNAdust1.06.tgz). The remaining reads were aligned to the human genome (hg19) using STAR (v2.5.1b) (Dobin et al, 2013) with the following parameters: --runThreadN 12 --outSAMtype BAM SortedByCoordinate --outFilterMismatchNoverReadLmax 0.04 --outFilterMultimapNmax 10. Mapping statistics can be found in Dataset EV7. TSSs were identified according to the protocol provided at http://fantom.gsc.riken.jp/5/sstar/Protocols:HeliScopeCAGE_read_alignment. Decomposition peak identification (https://github.com/hkawaji/dpi1/blob/master/identify_tss_peaks.sh) was used to identify tag clusters with default parameters but without decomposition. These peaks were used to identify bidirectional enhancers (https://github.com/anderssonrobin/enhancers/blob/master/scripts/bidir_enhancers).

The reads mapped to promoters (CAGE) and enhancers (NET-CAGE) were counted in a strand-specific manner with the UCSC software bigWigAverageOverBed (Kent et al, 2010), with the bigWig files produced from the decomposition peak identification as input. Lists of promoters and enhancers identified in the FANTOM5 project were obtained from https://fantom.gsc.riken.jp/5/data/. A prior count of 0.25 was added to the raw counts, which were then normalized based on the library size and converted to counts per million (CPM) using the R package edgeR (v3.40.2) (Robinson et al, 2010). Promoters with $\log_2$ [CPM] > $-2$ in at least one sample and enhancers with $\log_2$ [CPM] > $-2.5$ in at least one sample were retained. The list of tissue-specific enhancers was obtained from SlideBase (https://pressto.binf.ku.dk/) (Ienasescu et al, 2016). An analysis of variance (ANOVA)-like test was performed using the glmQLFTest function of edgeR. Peaks with a false-discovery rate <0.05 were considered significant and were classified into four clusters based on k-means clustering. Gene ontology (GO) term enrichment analysis was performed using the R package clusterProfiler (v4.7.1.3) (Wu et al, 2021). Motif enrichment was analyzed using the command findMotifsGenome.pl from HOMER (v4.11) (Heinz et al, 2010) with the options "-size -300,100 -mask" for promoters and "-size -400 -mask" for enhancers. All identified promoter and enhancer sequences were used as background, respectively. Motif activity was calculated using the scripts available at http://fantom.gsc.riken.jp/5/suppl/Alam_et_al_2020/data/MotifActivity (Alam et al, 2020). NEUROD1 ChIP-seq (GSM2432952) (Boulay et al, 2017a, 2017b) intensity around the enhancers was calculated and visualized using the computeMatrix and plotProfile commands of deepTools (v3.5.2) (Ramírez et al, 2016). Disease-associated SNPs were obtained from the NHGRI-EBI GWAS catalog (downloaded on March 15, 2024) (Buniello et al, 2019). GREGOR (v1.4.0) (Schmidt et al, 2015) was used to assess SNP enrichment within enhancers, with the following options: $r^2$ threshold = 0.7; LD window size = 1 Mb; minimum neighbor number = 500; and population = European. Randomly selected coding regions and genomic sequences were matched to the novel enhancers in both the number of regions (31,057) and length distribution. Random coding regions were selected from those annotated in GENCODE version 19 (Harrow et al, 2012), while random genomic sequences were selected from the whole genome, excluding telomeric and centromeric regions, using BEDtools (v2.31.1) (Quinlan and Hall, 2010). PhastCons scores were downloaded from the UCSC Genome Browser (https://hgdownload.cse.ucsc.edu/goldenpath/hg19/phastCons100way/) and calculated for these regions using the UCSC software bigWigAverageOverBed (Kent et al, 2010). Variant and allele frequency data were obtained from gnomAD (v2.1.1) (Karczewski et al, 2020), and SNPs overlapping with these regions were extracted using bcftools (v1.21) (Danecek et al, 2021). Nucleotide diversity ($\pi$) was calculated as 2 × allele frequency × (1 − allele frequency) for all SNPs overlapping these regions, and mean values for each of the 31,057 regions were compared. To avoid log 0, the smallest non-zero value was added to all values. Substantia nigra-specific eQTLs were obtained from the GTEx Portal GTEx Analysis v7 (Battle et al, 2017), and only SNPs with $P$-values < 0.05 were selected. These SNPs were intersected with the SNPs located in these regions using BEDtools.

## Capture Hi-C probe design

We used HiCapTools to design probes for selected enhancers and GWAS variants. To increase the number of GWAS variants overlapping with enhancers, we further identified bidirectional enhancers at each time point (shown as "overlapenh" in Dataset EV4). The following parameters were used for probe design: ProbeLength = 120, MinDistanceBetweenProbes = 1000, MinDistancebetweenFeatureandProbe = 300, MaxDistanceBetweenFeatureandProbetoTSS = 1200, MappabilityThreshold = 0.7, ClusterPromoters = 1200, and ExtentofRepeatOverlaps = 6. In total, we used interactions of 25,514 promoters or features for the analyses as well as 3000 negative control regions to calculate the background interaction frequency (Anil et al, 2018). The probe sequences can be found in Dataset EV5. We chose all the differentially expressed putative enhancers detected during LUHMES differentiation (2172). The part of the rest of the space is used to target all GWAS SNPs associated with neurodevelopmental diseases (NDD). The remaining space is used to target putative enhancers detected in LUHMES cells that overlap with SNPs that are in linkage disequilibrium with NDD-GWAS SNPs. The probe description in Dataset EV5 contains information

regarding their overlap with GWAS variants or those that are in LD with GWAS variants.

## Capture Hi-C sample preparations and sequencing

Approximately $3 \times 10^7$ LUHMES cells/sample were collected at the same time points used for CAGE and NET-CAGE analyses (Day 1, Day 3, and Day 6) following previously described protocols (Sahlén et al, 2015). Cells were washed with ice-cold PBS supplemented with cOmplete EDTA-free Protease Inhibitor Cocktail (Roche 4693159001; 1x) and fixed with formaldehyde (Thermo Fisher Scientific 28906; 1%) for 10 min at room temperature, with occasional flask swirling. The fixation reaction was stopped by adding glycine (0.125 M) for 5 min at room temperature, with rocking of the flask. The glycine solution was then removed before adding PBS with cOmplete EDTA-free Protease Inhibitor (1x). Cells were scraped off the flask and centrifuged at $300 \times g$ for 5 min at 4 °C.

The Hi-C reactions and sequence captures were performed as previously (Zhigulev and Sahlén, 2022). In brief, crosslinked cells were resuspended in 5 mL cell lysis buffer each, containing 10 mM Tris-HCl pH 8.0 (Sigma-Aldrich 93283-100 ML), 10 mM NaCl (Sigma-Aldrich S6546-1L), 0.2% Triton-X 100 (Sigma-Aldrich X100-5ML), 1x EDTA-free Protease inhibitor tablets (Sigma-Aldrich (Roche) 4693159001), incubated on ice for 15 min, centrifuged at $600 \times g$ for 10 min at 4 °C to collect nuclei. We resuspended each pellet in 550 μl 1x MboI FastDigest Buffer (Thermofisher FD0814), transferred 50 μl to a microtube, and kept at 4 °C as undigested control. We added 7.5 μl 20% SDS (Thermofisher AM9820, 0.3% final concentration (FC)) to each sample and incubated for 60 min shaking at 950 rpm, 37 °C, then quenched the reactions with 50.8 μl 20% Triton-X 100 (Sigma-Aldrich X100-5ML, 2% FC) and incubated for 60 min shaking at 950 rpm, 37 °C. We added 6 μl FastDigest MboI enzyme (Thermofisher FD0814) to each sample and incubated for 2 h shaking at 450 rpm, 37 °C, and incubated the samples at 75 °C for 10 min for inactivation. We added 1.2 μL of 10 mM dTTP (Thermofisher R0171), 1.2 μL of 10 mM dGTP (Thermofisher R0161), 1.2 μL of 10 mM dCTP (Thermofisher R0151), 30 μL biotin-14-dATP (Thermofisher 19524-016), and 1.2 μL of Klenow enzyme (Thermofisher EP0051, 10 U/μL, for end-repair), and incubated for 4 h at 23 °C shaking at 450 rpm for end-repair and later added 9.6 μL 0.5 M EDTA (Thermofisher PR-V4231), and incubated the sample at 75 °C for 10 min to inactivate the enzyme. The ligation reaction was set up by adding 150 μl of 10x FastDigest MboI buffer (Thermofisher FD0814), 15 μl of 100 mM ATP (Sigma-Aldrich A6559-25UMO), and 490 μl of the Hi-C sample from the previous step, final volume completed to 1.5 mL with nuclease-free water and 50 Weiss units of T4 DNA ligase (BioNordika (NEB) M0202L) was added and incubated at 16 °C overnight and 30 min at RT. Then, 9 μl proteinase K (Thermofisher EO0492) was added to Hi-C samples and incubated at 65 °C for 6 h. The DNA was isolated by adding equal volume Phenol:Chloroform:Isoamyl alcohol (25:24:1) sol (PCI) (Sigma-Aldrich 77617-100 ML) and collecting the upper layer. The DNA was precipitated by adding 0.1X v/v 3 M NaOAc (Sigma-Aldrich R1181) at pH 5.2 and 2.5X v/v 100% EtOH and 0.002X v/v μl glycogen (Thermofisher 10814-010) to each sample and incubated the samples at −20 °C overnight and DNA the pellet was washed with 200 μl 70% EtOH, pelleted at 16,000 × g for 5 min

at 4 °C and Hi-C samples where resuspended in 100 μl nuclease-free water. The Hi-C library was prepared as previously reported (Zhigulev and Sahlén, 2022). We used 3000 ng of Hi-C sample for each experiment. The adapter-ligated library was amplified using 8 cycles. Then 750 ng of each library was hybridized to the custom-designed sequence capture probes (Agilent Inc) using standard protocol from the vendor. The captured material was amplified 5 cycles. The finished libraries were pooled for sequencing on the NovaSeq 6000 Platform (151-10-10-151 and flowcell S4-300 v1.5).

## Capture Hi-C data processing

We first truncated Capture Hi-C reads using the HiCUP pipeline before mapping with bowtie2 (v2.2.9) (Langmead et al, 2009) to the hg19 genome assembly and removal of duplicates using the HiCUP *deduplicator* module (Wingett et al, 2015). The mapping statistics can be found in Dataset EV8. We then processed deduplicated files using HiCapTools to call interactions of targeted regions (Anil et al, 2018). We used the interaction profile of 3000 negative control regions (regions with no annotated promoter or enhancer activity, Dataset EV5) to calculate the true *P*-values for each interaction. We required each interaction to have at least five supporting pairs and to have a Bonferroni-adjusted *P*-value of 0.1 in both replicates. We removed interacting regions within blacklisted and high-signal regions (Amemiya et al, 2019). We compared the interaction frequency of targeted enhancers to that of negative controls using a two-sample test for equality of proportions with continuity correction. Genome locus figures showing interactions were plotted using the R package plotgardner (v1.8.2) (Kramer et al, 2022).

## Assessment of enhancer potential of Capture Hi-C promoter-interacting regions

We only considered distal regions not targeted by any probes for enrichments to avoid bias. We downloaded two datasets from ChIP-Atlas (Zou et al, 2024) that contain H3K27Ac and TF binding signals from neural datasets (His.Neu.10.H3K27ac.AllCell.bed and Oth.Neu.10.AllAg.AllCell.bed), using an FDR cutoff of 0.01 for peak calling and including only replicated peaks. We also downloaded cCRE (candidate cis-Regulatory Element) regions from the UCSC genome table browser. We performed overlaps on merged datasets to avoid multiple counts of the same overlaps. We used the R package valr (v0.8.1) (Riemondy et al, 2017) to perform Fisher's exact test and calculate the JI between the sets.

## Assessment of interactome changes for each target

To this aim, we generated interaction networks as follows: each interacting promoter or enhancer was considered a node in the network, and two nodes were connected if they had a significant interaction in at least one replicate. We used the R package igraph (v1.4.2) (Csárdi and Nepusz, 2006) to generate the networks. We then created a summary of interactions for each node by summing its interactions.

To calculate the JI and OCE of a node between the two networks (A, B), we used the following formulas:

$$JI_x = \frac{A \cap B}{A \cup B}$$

$$OCE_x = \frac{A \cap B}{\min(A, B)}$$

where $A$ and $B$ denote the number of neighbors of node $x$ in networks A and B, respectively.

Importantly, the network is generated using non-replicated interactions (i.e., those that are called interactions in one of the states/replicates); therefore, the JI and OCE parameters reflect the amount of novel (gained) or lost interactions. Note that a node with a JI value less than one can have an OCE value equal to one. This is because OCE requires that the number of common nodes is greater than the number of nodes in the smaller network. For example, a node with one connection at one timepoint could lose its connection upon differentiation and not gain any additional connections. The JI would be zero (since the denominator is 1), but the OCE would be one since the network with the minimum number of nodes is zero (i.e., the denominator). Therefore, OCE is a stricter measure of connectivity change than JI.

### Nearest gene localization

We used R packages GenomicFeatures (v1.57.0) (Lawrence et al, 2013), RMariaDB (v1.3.2) (https://cran.r-project.org/package=RMariaDB), and ChIPpeakanno (v3.39.2) (Zhu et al, 2010) to map the distal interacting regions to their nearest genes. The function "annotatePeakInBatch" was used to find the nearest genes and compare the HGNC symbol of the assigned nearest gene and interacting gene. Therefore, only the promoters with canonical names (40,972 out of 53,761) could be included in the analysis.

### gRNA design and PCR assembly of gRNA cassettes

gRNA sequences were designed using the Benchling CRISPR tool (https://benchling.com/), targeting proximal promoters (−400 to −100 bp from TSSs) of genes of interest and inside the selected enhancer sequences. Possible guides were selected according to their off-target scores and positions (see Dataset EV6 for details). gRNA transcriptional units (gRNA-PCRs) were prepared by PCR amplification with Phusion Polymerase (Thermo Fisher Scientific F350), using the U6 promoter as a template, and terminator PCR products were amplified together with a gRNA sequence containing oligo to bridge the gap (Balboa et al, 2015; Weltner and Trokovic, 2021). PCR reactions contained 50 pmol forward and reverse primers, 2 pmol guide oligos, 5 ng U6 promoter, and 5 ng terminator PCR product in a total reaction volume of 100 μL. The PCR reaction program consisted of 98 °C for 10 s, 56 °C for 30 s, and 72 °C for 12 s for 35 cycles. Amplified gRNA-PCRs were purified and transfected into hTERT-RPE1 cells.

### Cell transfection

hTERT-RPE1 cells were seeded on treated 12-well tissue culture plates 1 day prior to transfection ($6.5 \times 10^4$ cells/well). Cells were transfected using 2 μg of Lipofectamine 3000 transfection reagent (Invitrogen L3000001) in Opti-MEM medium (Gibco 31985062) with 1 μg of dCas9 activator-encoding plasmid (CAG-dCasVPN192-T2A-EGFP) and 400 ng of gRNA-PCR product or

TdT gRNA. Four hours after transfection, the medium was changed to standard hTERT-RPE1 cell culture medium and cells were cultured for 24 h post-transfection, after which samples were collected for qRT-PCR. We conducted three independent experiments for each gene.

### qRT-PCR

Total RNA was isolated using the RNeasy Mini Kit (Qiagen 74104), and RNA quality and concentration were measured using a NanoDrop 8000 Spectrophotometer (Thermo Fisher Scientific ND-8000-GL). Starting with 0.5 μg of high-quality total RNA, cDNA was prepared using the RevertAid H Minus First Strand cDNA Synthesis Kit (Thermo Fisher Scientific K1631), using a 1:1 mix of random hexamers and anchored oligo dT primers. cDNAs were subsequently diluted three times, and the relative abundance of each mRNA species was assessed by qRT-PCR using the 7500 Fast Real-Time PCR System (Applied Biosystems), employing 2 μL of the diluted samples in a final volume of 15 μL. The primers used for the amplification are listed in Dataset EV9. All the data were normalized to the endogenous reference gene *GAPDH*. The rate of increased expression was calculated using the $2^{-\Delta\Delta Ct}$ method (Livak and Schmittgen, 2001), and the cells transfected with dCas9 activator together with TdT targeting guides were used as controls.

### Statistical analyses

Differences in mRNA levels were assessed by a one-way ANOVA, followed by Dunnett's multiple comparisons test. *P*-values < 0.05 were considered to be statistically significant. No sample size estimate was performed, but sample size was selected based on previous studies. No blinding was done in this study.

## Data availability

CAGE and NET-CAGE data: ArrayExpress database at EMBL-EBI 'E-MTAB-13588'. Capture Hi-C data: ArrayExpress database at EMBL-EBI 'E-MTAB-13665'.

The source data of this paper are collected in the following database record: biostudies:S-SCDT-10_1038-S44319-025-00372-1.

## Peer review information

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

## Acknowledgements

This work was supported by the Japan Society for the Promotion of Science (JSPS) Grants-in-Aid for Scientific Research (KAKENHI) grant number 22K20622 (MY), Carl Tryggers Stiftelse (CTS) grant agreement No. CTS-19311 (PSa), Swedish Research Council (PSw, JK), Swedish Brain Foundation (PSw, JK), Sigrid Jusélius Foundation (JK), Jane and Aatos Erkko Foundation (JK), KI Strategic Neuroscience Program (PSw), Torsten Söderberg Foundation (PSw), Olle Engkvist Foundation (PSw, MP), Åhlén Foundation (PSw), Karolinska Institute PhD student (KID) Fellowship (AC, HL), JSPS Postdoctoral Fellowship for Research in Japan (SB), and Chinese Scholarship Council (CSC) PhD student Fellowship (HL). Some of the computations were enabled by resources from project NAISS 2023/22-408 provided by the National Academic Infrastructure for Supercomputing in Sweden (NAISS) at UPPMAX, funded by the Swedish Research Council through grant agreement no. 2022-06725. CAGE and NET-CAGE libraries were sequenced at DNAFORM (Yokohama, Japan).

## Author contributions

**Masahito Yoshihara**: Conceptualization; Data curation; Software; Formal analysis; Funding acquisition; Investigation; Visualization; Methodology; Writing—original draft; Writing—review and editing. **Andrea Coschiera**: Resources; Validation; Investigation; Visualization; Methodology; Writing—original draft; Writing—review and editing. **Jörg A Bachmann**: Investigation; Methodology; Writing—original draft; Writing—review and editing. **Mariangela Pucci**: Validation; Investigation; Writing—review and editing. **Haonan Li**: Validation; Investigation; Writing—review and editing. **Shruti Bhagat**: Methodology; Writing—review and editing. **Yasuhiro Murakawa**: Methodology; Writing—review and editing. **Jere Weltner**: Methodology; Writing—review and editing. **Eeva-Mari Jouhilahti**: Methodology; Writing—review and editing. **Peter Swoboda**: Conceptualization; Resources; Supervision; Funding acquisition; Writing—original draft; Project administration; Writing—review and editing. **Pelin Sahlén**: Data curation; Software; Formal analysis; Supervision; Funding acquisition; Visualization; Writing—original draft; Project administration; Writing—review and editing. **Juha Kere**: Conceptualization; Supervision; Funding acquisition; Writing—original draft; Project administration; Writing—review and editing.

Source data underlying figure panels in this paper may have individual authorship assigned. Where available, figure panel/source data authorship is listed in the following database record: biostudies:S-SCDT-10_1038-S44319-025-00372-1.

## Funding

## Disclosure and competing interests statement

YM is an inventor on a patent related to the NET-CAGE technology. Other authors declare no competing interests.

# Expanded View Figures

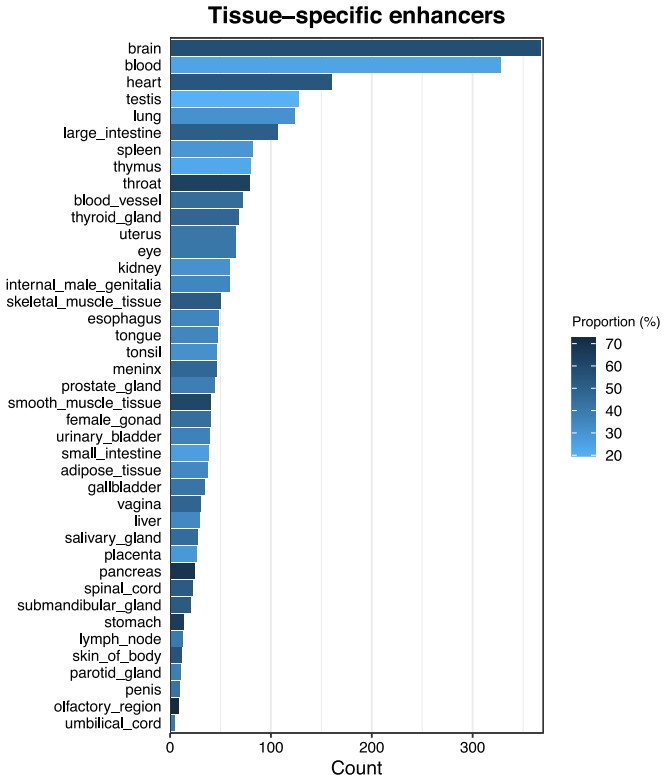

**Figure EV1.  Tissue-specific enhancers overlapping with the putative enhancers expressed in LUHMES.**

Bar plot showing the number of tissue-specific enhancers identified in the FANTOM5 project that overlap with the putative enhancers expressed in LUHMES. Colors indicate the proportion of specific enhancers in each tissue that overlap with the putative enhancers expressed in LUHMES. Tissues are sorted based on the number of overlapping enhancers.

**A**

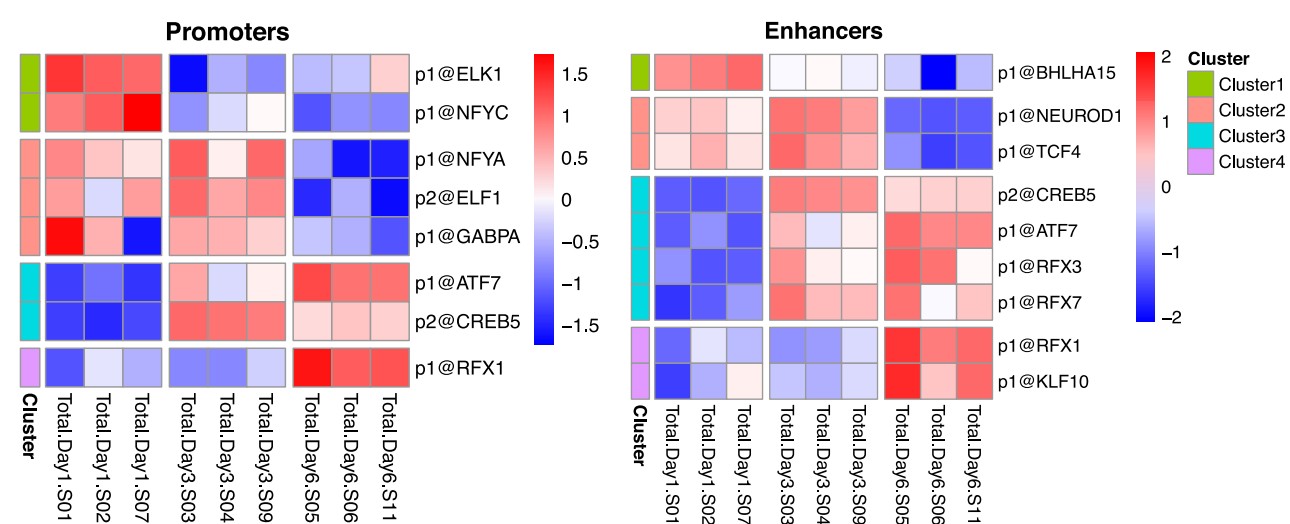

**B**

## Motif enrichment analysis of novel enhancers (31,057)

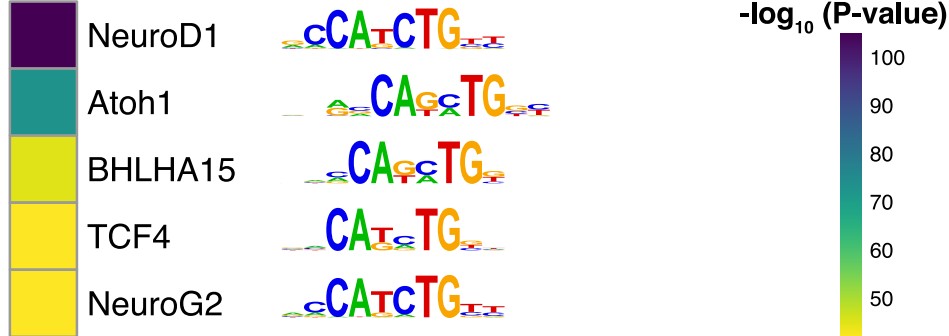

**C**

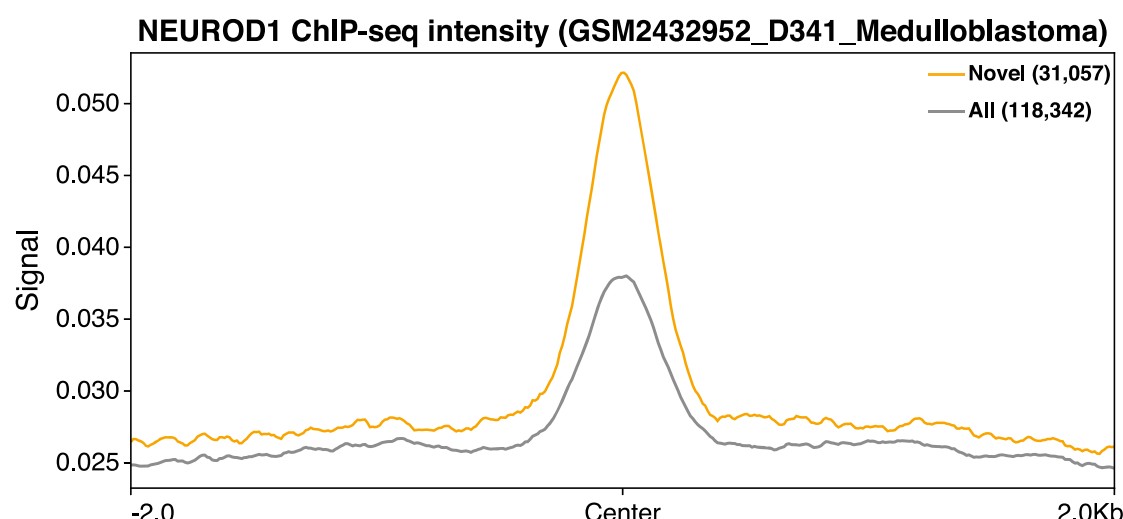

◀ **Figure EV2. Transcription factors likely to function during LUHMES neuronal differentiation.**

(A) Heatmaps showing the expression levels of the promoters of transcription factors whose DNA binding motifs are enriched in the promoters and enhancers of the same cluster. Promoters of transcription factors shown in Fig. 3A, B were investigated. (B) Heatmap showing the enrichment of transcription factor binding motifs in the novel putative enhancer regions (31,057) in LUHMES. *P*-values were calculated using the binomial test. (C) NEUROD1 ChIP-seq intensity around the novel putative enhancer regions (31,057) and all identified putative enhancers (118,342) using a publicly available dataset.

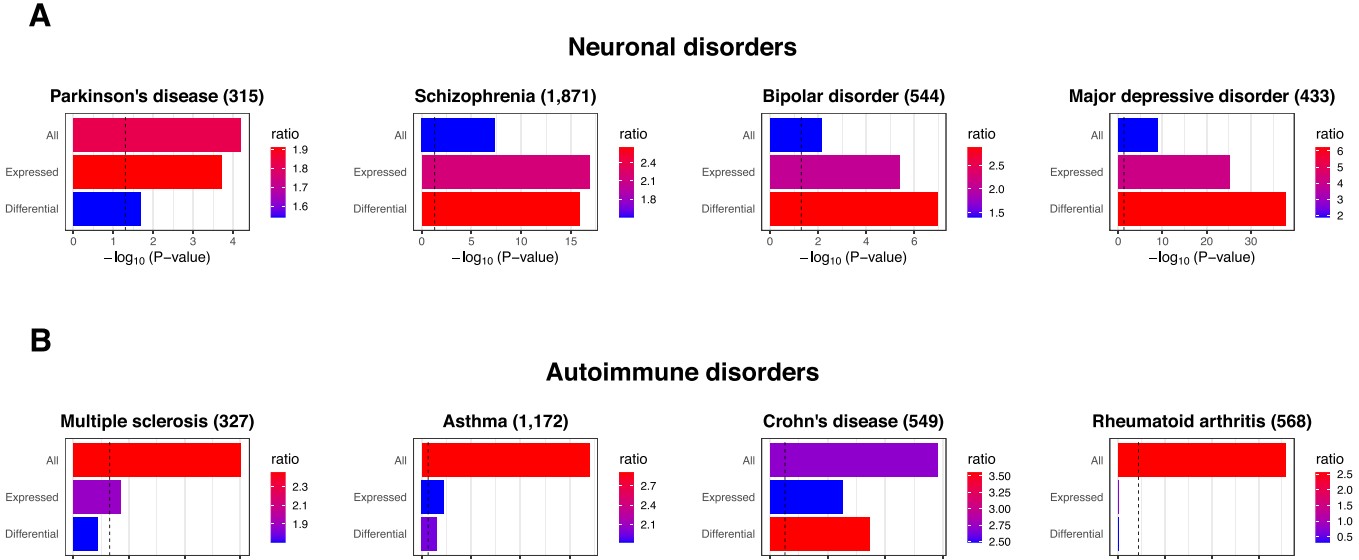

**Figure EV3.  Enrichment of neuronal disorder-associated GWAS SNPs within promoter regions.**

Enrichment of GWAS SNPs associated with neuronal (**A**) or autoimmune disorders (**B**) in all promoter regions (All; 184,827), promoter regions expressed in LUHMES (Expressed; 52,076), and promoter regions differentially expressed in LUHMES (Differential; 21,907). *P*-values were calculated using the permutation test. The dashed lines represent $P = 0.05$. Numbers in parentheses indicate the number of SNPs for each disorder.

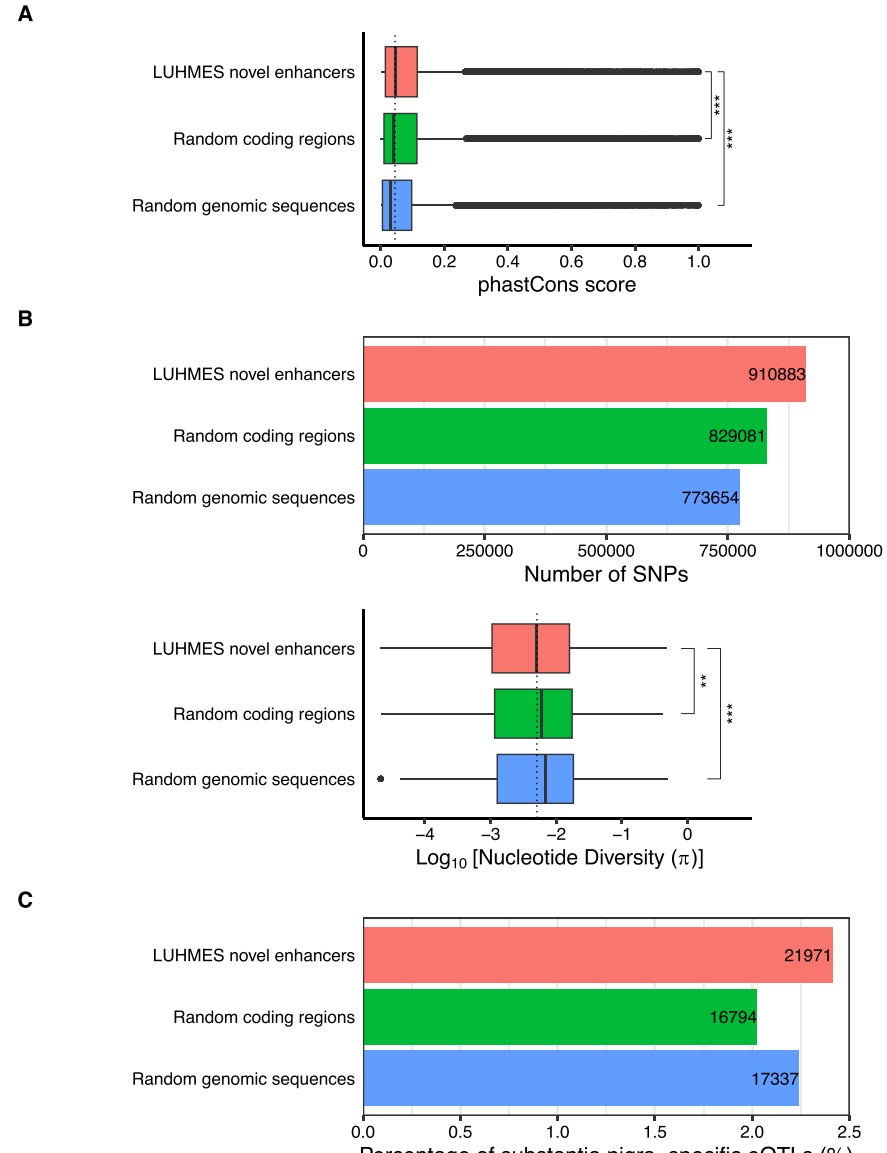

**Figure EV4. Conservation and variant analysis of the novel putative enhancer regions.**

(A) Box plots showing the distribution of phastCons scores for the novel putative enhancer regions, random coding regions, and random genomic sequences. Each group contains 31,057 regions. The dotted vertical line indicates the median value of the novel putative enhancer regions. ***$P < 2.2 \times 10^{-16}$; Wilcoxon rank sum test. (B) Top: Bar plots showing the number of SNPs overlapping with the novel putative enhancer regions, random coding regions, and random genomic sequences. Bottom: Box plots showing the distribution of $\log_{10}$ [nucleotide diversity ($\pi$)] for SNPs overlapping these regions. The dotted vertical line indicates the median value of the novel putative enhancer regions. **$P = 1.1 \times 10^{-14}$, ***$P < 2.2 \times 10^{-16}$; Wilcoxon rank sum test. Center lines in the box plots represent the medians. Box limits indicate 25th and 75th percentiles, while whiskers extend to 1.5 times the interquartile range (IQR) beyond the box limits. Data points outside this range are shown as outliers. (C) Bar plots showing the percentage of SNPs overlapping with the substantia nigra-specific eQTLs in each region. The numbers on the bars indicate the number of SNPs overlapping with these eQTLs. Significant enrichment was observed in the novel putative enhancers compared to the random coding regions ($P < 2.2 \times 10^{-16}$; Fisher's exact test) and the random genomic sequences ($P < 2.2 \times 10^{-16}$).

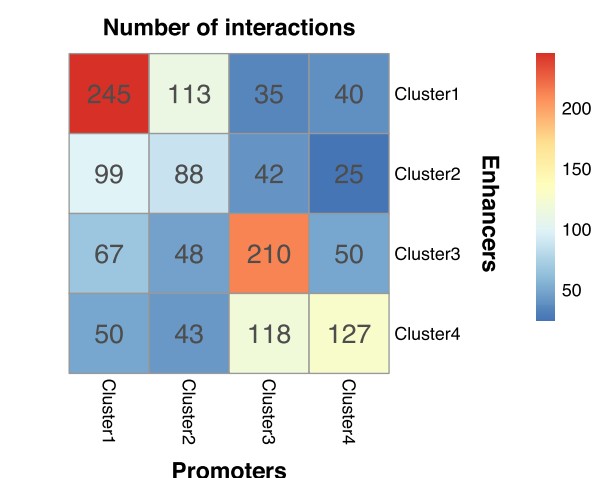

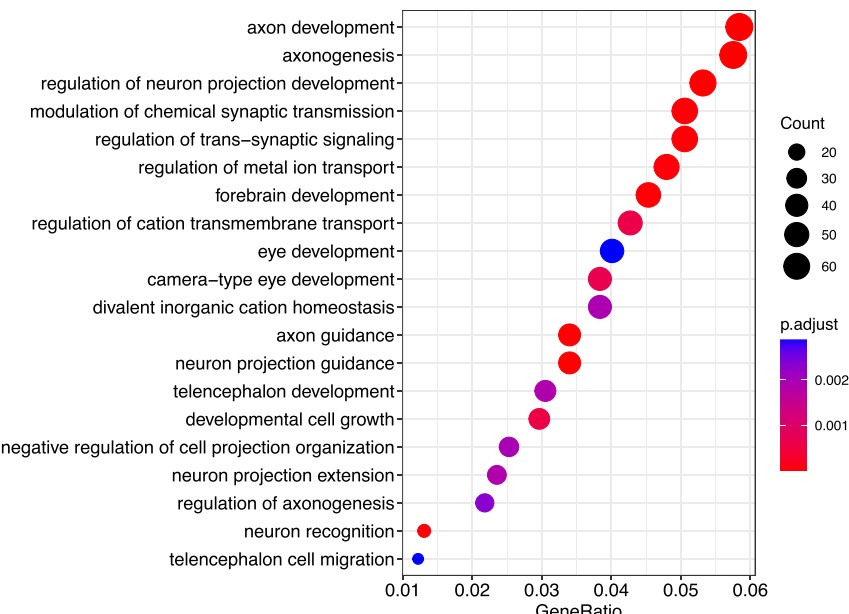

**Figure EV5.  Characterization of enhancer–promoter interactions.**

(**A**) Heatmap showing the number of interactions between enhancers (row) and promoters (column) belonging to each cluster. (**B**) Gene Ontology (GO) term enrichment analysis of the 1243 target genes of the enhancers identified in LUHMES. *P*-values were calculated using the hypergeometric test and adjusted using the Benjamini-Hochberg method.

