## [Peer Review File · EMBO Reports]

Transcriptional enhancers in human neuronal differentiation provide clues to neuronal disorders

Masahito Yoshihara, Andrea Coschiera, Jörg Bachmann, Mariangela Pucci, Haonan Li, Shruti Bhagat, Yasuhiro Murakawa, Jere Weltner, Eeva-Mari Jouhilahti, Peter Swoboda, Pelin Sahlén, and Juha Kere

Corresponding authors: Juha Kere (juha.kere@ki.se), Peter Swoboda (peter.swoboda@ki.se), Pelin Sahlén (pelinak@kth.se)

Review Timeline:

Submission Date:	31st Aug 24
Editorial Decision:	8th Oct 24
Revision Received:	8th Nov 24
Editorial Decision:	13th Dec 24
Revision Received:	28th Dec 24
Accepted:	9th Jan 25

Editor: Esther Schnapp

Transaction Report:

Dear Prof. Kere,

Thank you for the submission of your manuscript to EMBO reports. I could only secure 2 referees for your ms by now, and we have now received their comments that are pasted below. Given that both referees are in fair agreement, I am making a decision on your study now based on the two reports we have.

As you will see, the referees acknowledge that the findings are potentially interesting. However, they also have some suggestions for how the study could be improved and I think all suggestions are good and should be addressed. Please let me know in case you disagree and we can discuss the exact revision requirements further, also in a video chat, if you like.

I would thus like to invite you to revise your manuscript with the understanding that the referee concerns must be fully addressed and their suggestions taken on board. Please address all referee concerns in a complete point-by-point response. Acceptance of the manuscript will depend on a positive outcome of a second round of review. It is EMBO reports policy to allow a single round of major revision only and acceptance or rejection of the manuscript will therefore depend on the completeness of your responses included in the next, final version of the manuscript.

We realize that it is difficult to revise to a specific deadline. In the interest of protecting the conceptual advance provided by the work, we recommend a revision within 3 months (8th Jan 2025). Please discuss the revision progress ahead of this time with the editor if you require more time to complete the revisions.

- 1) A data availability section providing access to data deposited in public databases is missing. If you have not deposited any data, please add a sentence to the data availability section that explains that.
- 2) Your manuscript contains statistics and error bars based on $n=2$. Please use scatter blots in these cases. No statistics should be calculated if $n=2$.

3) We replaced Supplementary Information with Expanded View (EV) Figures and Tables that are collapsible/expandable online. A maximum of 5 EV Figures can be typeset. EV Figures should be cited as 'Figure EV1, Figure EV2' etc... in the text and their respective legends should be included in the main text after the legends of regular figures.

5) a complete author checklist, which you can download from our author guidelines <https://www.embopress.org/page/journal/14693178/authorguide>. Please insert information in the checklist that is also reflected in the manuscript. The completed author checklist will also be part of the RPF.

6) Please note that all corresponding authors are required to supply an ORCID ID for their name upon submission of a revised manuscript (<https://orcid.org/>). Please find instructions on how to link your ORCID ID to your account in our manuscript tracking system in our Author guidelines <https://www.embopress.org/page/journal/14693178/authorguide#authorshipguidelines>

12) All Materials and Methods need to be described in the main text using our 'Structured Methods' format, which is required for all research articles. According to this format, the Methods section includes a Reagents and Tools Table (listing key reagents, experimental models, software and relevant equipment and including their sources and relevant identifiers) followed by a Methods and Protocols section describing the methods using a step-by-step protocol format. The aim is to facilitate adoption of the methodologies across labs. More information on how to adhere to this format as well as a downloadable template (.docx) for the Reagents and Tools Table can be found in our author guidelines: <https://www.embopress.org/page/journal/14693178/authorguide#structuredmethods>.

An example of a Method paper with Structured Methods can be found here: <https://www.embopress.org/doi/full/10.1038/s44320-024-00037-6#sec-4>

I look forward to seeing a revised form of your manuscript when it is ready.

Referee #1:

This manuscript applied advanced genomic technologies like NET-CAGE and Capture Hi-C to study the role of non-coding GWAS variants in neuropsychiatric disorders, mainly using the LUHMES human mesencephalic neuronal cell line as a model system. While this study is a significant advancement in linking non-coding variants to their potential regulatory roles in neuropsychiatric disorders, two main aspects should be addressed before acceptance.

1. Presentation: The manuscript's figures are underwhelming in terms of delivering the main conclusions drawn from the high-dimensional genomic data. More effective data visualization would enhance the communication of their findings.

Fig 1C- Clearly describe the specific criteria used for filtering.

Fig 2A,C - The labels under the heat map need to be more precise (e.g., what does the total mean represent in Figure 2A? Are the expression data from NET-CAGE? Are S01-S11 sample numbers?). Include appropriate scale information.

Fig 2D and the results of Capture Hi-C should be compared later.

Fig 6A - The main message is unclear and needs clarification.

Fig 6C - Present the NET-CAGE results as a genome browser view, alongside Capture Hi-C results, to enable direct comparison.

Fig 7 - Display all data points on the bar graphs to provide a more precise overview of data distribution.

2. Neurodevelopmental cellular model: Using LUHMES cells to study neurodevelopmental enhancers and perform genomic analyses is reasonable. Interestingly, the validation of enhancers and linked genes in PRE cells was successful, suggesting the identification of general early-stage pan-neuronal enhancers, including some still active in dopaminergic neurons. This could be related to the early neurodifferentiation stages in LUHMES cells. The authors should discuss this broader enhancer functionality. Perhaps a significant Schizophrenia GWAS link is associated with this very early stage of neurodifferentiation and broader neuronal enhancers instead of a dopaminergic neuronal link. Additional neuronal cell types should be examined in linking enhancers to genes to argue for cell-type-specific effects.

Referee #2:

The genetic variability of enhancers is thought to be a rich genomic space for causative variants in mendelian and non-mendelian disorders and traits. Thus the description of the entire catalogue of enhancers in the genome is a necessary infrastructure discovery. This study substantially contributes to this objective. The methodology is state of the art, and the conclusions are reasonable. The results of this study provide a rich source of new enhancers and expands the genomic space of these functional genomic elements.

Some suggestions/requests for improvement:

1. Please provide information of the false positive and false negative likely-enhancers.

2. What is the conservation (in mouse) of the novel enhancer sequences described in this study?

3. What is the polymorphic variability of the likely-enhancers in the population? Are these enhancers relatively intolerant to variation? Data from the UK Biobank or other sequencing databases could be used.

4. What is the fraction of eQTLs among the SNPs in them? Is there an enrichment in eQTLs within these enhancer sequences?

5. The novel enhancers described will be very valuable to the scientific community. Therefore special attention needs to be paid for the availability of their genomic coordinates. Could the authors include the enhancers in a custom track of the UCSC browser, or provide a list of genomic coordinates available to the community?

MS ID# EMBOR-2024-60184V1

Yoshihara et al. "Transcriptional enhancers in human neuronal differentiation provide clues to neuronal disorders"

The point-by-point response to the Reviewers' comments

We thank the Reviewers for providing helpful and constructive comments. We are pleased that this process has resulted in a significantly improved manuscript. We hope that you will find our information helpful. Thank you once again for considering this work.

Below is the point-by-point response to the Reviewers' comments (*comments copied in italics*) followed by our response:

Referee #1:

General comment: *This manuscript applied advanced genomic technologies like NET-CAGE and Capture Hi-C to study the role of non-coding GWAS variants in neuropsychiatric disorders, mainly using the LUHMES human mesencephalic neuronal cell line as a model system. While this study is a significant advancement in linking non-coding variants to their potential regulatory roles in neuropsychiatric disorders, two main aspects should be addressed before acceptance.*

Response: We thank the Reviewer's positive comments and kind suggestions.

Specific comment 1-1: *Fig 1C- Clearly describe the specific criteria used for filtering.*

Response: We apologize for the lack of clarity regarding the specific filtering criteria. As described in the Methods section, enhancers with \log_2 [CPM] > -2.5 in at least one sample were retained, following the criteria used in the NET-CAGE development paper (Hirabayashi *et al*, 2019). We have added this description to the legend of **Fig. 1D** in the revised manuscript.

Specific comment 1-2: *Fig 2A,C - The labels under the heat map need to be more precise (e.g., what does the total mean represent in Figure 2A? Are the expression data from NET-CAGE? Are S01-S11 sample numbers?). Include appropriate scale information.*

Response: We apologize for the confusion caused by the lack of information. The labels under the heat map indicate the sample names. 'Total' refers to total RNA measured by CAGE, while 'Nascent' refers to nascent RNA measured by NET-CAGE. S01–S11 represent sample numbers, as the Reviewer pointed out. We have added this information to the legends of **Fig. 2A and C** in the revised manuscript. We further included scale information (Z-score) in both **Fig. 2A and C**.

Specific comment 1-3: *Fig 2D and the results of Capture Hi-C should be compared later.*

Response: We greatly appreciate the Reviewer's suggestion. Based on the Capture Hi-C results, we examined the number of interactions between enhancers and promoters belonging to each cluster and summarized these interactions as a heatmap in **Fig. EV5A** of the revised manuscript. We observed that upregulated enhancers tended to interact with upregulated promoters, while downregulated enhancers tended to interact with downregulated promoters. In the Results section, we have added: "*Except for cluster 2, enhancers most frequently interacted with promoters belonging to the same cluster (Fig. EV5A). Enhancers in cluster 2 interacted most frequently with promoters in cluster 1, both of which showed a downregulation pattern.*"

Specific comment 1-4: *Fig 6A - The main message is unclear and needs clarification.*

Response: We apologize for the lack of clarity. In the revised manuscript, we have rephrased the sentence in the Results section as follows: "*the expression levels of interacting promoters and enhancers in a dynamic manner (i.e., the interaction was observed only at a single time point), showed the strongest positive correlation at the time point when the interaction between them was observed (Fig. 6A).*"

Specific comment 1-5: *Fig 6C - Present the NET-CAGE results as a genome browser view, alongside Capture Hi-C results, to enable direct comparison.*

Response: We have already displayed the NET-CAGE results as the 'DE Enhancers' track, which shows the differentially expressed enhancers detected by NET-CAGE. Displaying all enhancers detected by NET-CAGE might, in our opinion, make the plot too crowded.

Specific comment 1-6: *Display all data points on the bar graphs to provide a more precise overview of data distribution.*

Response: We appreciate the Reviewer's valuable suggestion. In **Fig. 7** of the revised manuscript, we displayed all data points on the bar graphs to provide a clearer overview of the data distribution.

Specific comment 2: *Neurodevelopmental cellular model: Using LUHMES cells to study neurodevelopmental enhancers and perform genomic analyses is reasonable. Interestingly, the validation of enhancers and linked genes in PRE cells was successful, suggesting the identification of general early-stage pan-neuronal enhancers, including some still active in dopaminergic neurons. This could be related to the early neurodifferentiation stages in LUHMES cells. The authors should discuss this broader enhancer functionality. Perhaps a significant Schizophrenia GWAS link is associated with this very early stage of neurodifferentiation and broader neuronal enhancers instead of a dopaminergic neuronal link. Additional neuronal cell types should be examined in linking enhancers to genes to argue for cell-type-specific effects.*

Response: We appreciate the Reviewer's very insightful comment. It is definitely possible that the significant association of the newly identified enhancers does not just arise from the dopaminergic differentiation, but indeed from more general early pan-neuronal enhancers and regulatory mechanisms, their common role supported by the functional replication in RPE cells. We agree

with the Reviewer in that arguing cell-type (in this case, dopaminergic) specificity would require more comprehensive neuronal cell comparisons. Such experiments are out of the scope and reach of this study. Therefore, we have modified the Discussion to emphasize the point raised by the Reviewer and soften the statement about cell-type specificity (Discussion, 3rd paragraph): “As many of the enhancers identified here may have more general neuronal roles, we cannot suggest a direct link between dopaminergic regulation and any of the disease associations.”

Referee #2:

General comment: *The genetic variability of enhancers is thought to be a rich genomic space for causative variants in mendelian and non-mendelian disorders and traits. Thus the description of the entire catalogue of enhancers in the genome is a necessary infrastructure discovery. This study substantially contributes to this objective. The methodology is state of the art, and the conclusions are reasonable. The results of this study provide a rich source of new enhancers and expands the genomic space of these functional genomic elements.*

Response: We sincerely appreciate the Reviewer’s thoughtful and positive comments.

Specific comment 1: *Please provide information of the false positive and false negative likely-enhancers.*

Response: As we described in the Discussion section, each enhancer detection method has its own advantages and limitations, and so does NET-CAGE. In the revised manuscript, we have added the following: “Nevertheless, false positives may arise due to transcriptional noise or bidirectional transcription from insulators (Melgar et al, 2011) and accessible DNA (Young et al, 2017), while false negatives may result from eRNAs expressed at very low levels below detection thresholds or from enhancers that do not produce eRNAs (Catarino & Stark, 2018).”

Specific comment 2: *What is the conservation (in mouse) of the novel enhancer sequences described in this study?*

Response: We appreciate the Reviewer's valuable suggestion. We investigated the phastCons scores for the novel enhancers, as well as randomly selected coding regions and genomic sequences. The randomly selected coding regions and genomic sequences were matched to the novel enhancers in both the number of regions (31,057) and length distribution. We found that the phastCons scores for the novel enhancer sequences were significantly higher than those for randomly selected coding regions and genomic sequences, indicating that the novel enhancer sequences are highly conserved. The result is shown in Fig. **EV4A** of the revised manuscript.

Specific comment 3: *What is the polymorphic variability of the likely-enhancers in the population? Are these enhancers relatively intolerant to variation? Data from the UK Biobank or other sequencing databases could be used.*

Response: We are grateful to the Reviewer for this insightful suggestion. Using variant and allele frequency data obtained from gnomAD, we observed a higher number of SNPs within the novel enhancer regions compared to the two random sets, indicating considerable genetic variability. However, when evaluating nucleotide diversity for SNPs within these regions, we found that the novel enhancer regions displayed significantly lower diversity than the random sets. This suggests that the novel enhancer regions may be relatively intolerant to variation and likely subject to selective pressure. These results are now shown in Fig. **EV4B** of the revised manuscript.

Specific comment 4: *What is the fraction of eQTLs among the SNPs in them? Is there an enrichment in eQTLs within these enhancer sequences?*

Response: We thank the Reviewer for this suggestion. Since LUHMES is a midbrain-derived dopaminergic neuronal cell line, we examined the fraction of substantia nigra-specific eQTLs among the SNPs within the novel enhancer regions and the two random sets. We observed a significant enrichment of substantia nigra-specific eQTLs in the novel enhancer regions, suggesting that these enhancers may play functionally significant roles in midbrain dopaminergic neurons. The result is shown in Fig. **EV4C** of the revised manuscript.

Specific comment 5: *The novel enhancers described will be very valuable to the scientific community. Therefore special attention needs to be paid for the availability of their genomic coordinates. Could the authors include the enhancers in a custom track of the UCSC browser, or provide a list of genomic coordinates available to the community?*

Response: We wish to thank the Reviewer for this comment. We have already provided the list of genomic coordinates for all the identified enhancers as **Dataset EV1**. Readers can easily convert this to a BED file and upload it to the UCSC browser as a custom track.

Dear Prof. Kere,

Thank you for the submission of your revised manuscript. We have now received the enclosed reports from the referees and I am happy to say that both support its publication now. Only a few editorial requests will need to be addressed before we can proceed with the official acceptance of your manuscript:

- Please move the Data Availability Section to before the Acknowledgments.
- Please remove the author credits from the ms file. All credits need to be entered during online ms submission.
- In the author checklist, the section on Statistics has not been filled in. Please send us a new, completed checklist.
- The funding info in the ms and in our online submission system must be identical. This last part of the Acknowledgements needs to be entered in eJP too: Some of the computations were enabled by resources from project NAISS 2023/22-408 provided by the National Academic Infrastructure for Supercomputing in Sweden (NAISS) at UPPMAX, funded by the Swedish Research Council through grant agreement no. 2022-06725.
- Your Appendix file has suppl methods. This should only be the case if the methods exclusively refer to the figures shown in the Appendix. If these methods are also relevant for the main or EV figures, please move these methods to the Method section in the main ms file.

Figure Legends - Comments

- Please note that the exact p values are not provided in the legends of figures EV 4a-c.
- Please indicate the statistical test used for data analysis in the legends of figures 2b; 3a-b; 4a-b; 6b; EV 2b; EV 3a-b; EV 5b.
- Please note that the box plots need to be defined in terms of minima, maxima, centre, bounds of box and whiskers, and percentile in the legends of figures EV 4a-b.
- Please note that information related to n is missing in the legends of figures EV 4a-b.

I would like to suggest some minor changes to the abstract that needs to be written in present tense. Please let me know whether you agree with the following:

Genome-wide association studies (GWASs) have identified thousands of variants associated with complex phenotypes, including neuropsychiatric disorders. To better understand their pathogenesis, it is necessary to identify the functional roles of these variants, which are largely located in non-coding DNA regions. Here, we employ a human mesencephalic neuronal cell differentiation model, LUHMES, with sensitive and high-resolution methods to discover enhancers (NET-CAGE), perform DNA conformation analysis (Capture Hi-C) to link enhancers to their target genes, and finally validate selected interactions. We expand the number of known enhancers active in differentiating human LUHMES neurons to 47.350, and find overlap with GWAS variants for Parkinson's disease and schizophrenia. Our findings reveal a fine-tuned regulation of human neuronal differentiation, even between adjacent developmental stages; provide a valuable resource for further studies on neuronal development, regulation, and disorders; and emphasize the importance of exploring the vast regulatory potential of non-coding DNA and enhancers.

EMBO press papers are accompanied online by A) a short (1-2 sentences) summary of the findings and their significance, B) 2-3 bullet points highlighting key results and C) a synopsis image that is exactly 550 pixels wide and 200-600 pixels high (the height is variable). The synopsis image should provide a sketch of the major findings, like a graphical abstract. Please note that text needs to be readable at the final size. Please send us this information along with the final manuscript.

Referee #1:

I appreciate the authors' efforts in addressing my points. They provided satisfactory responses to all my comments.

Referee #2:

The authors have successfully addressed the concerns and criticisms. Figures EV4 A, B, and C are a welcome addition to the manuscript.

The authors addressed the remaining editorial issues.

Prof. Juha Kere
Karolinska Institutet
Sweden

Dear Prof. Kere,

I am very pleased to accept your manuscript for publication in the next available issue of EMBO reports. Thank you for your contribution to our journal.

Yours sincerely,
